# Cross-modality synthesis of EM time series and live fluorescence imaging

**Anthony Santella[1†], Irina Kolotuev[2\*†], Caroline Kizilyaprak[2], Zhirong Bao[3\*]**

[1]Molecular Cytology Core, Memorial Sloan Kettering Cancer Center, New York, United States; [2]Electron Microscopy Facility, University of Lausanne, Lausanne, Switzerland; [3]Developmental Biology Program, Memorial Sloan Kettering Cancer Center, New York, United States

**Abstract** Analyses across imaging modalities allow the integration of complementary spatiotemporal information about brain development, structure, and function. However, systematic atlasing across modalities is limited by challenges to effective image alignment. We combine highly spatially resolved electron microscopy (EM) and highly temporally resolved time-lapse fluorescence microscopy (FM) to examine the emergence of a complex nervous system in *Caenorhabditis elegans* embryogenesis. We generate an EM time series at four classic developmental stages and create a landmark-based co-optimization algorithm for cross-modality image alignment, which handles developmental heterochrony among datasets to achieve accurate single-cell level alignment. Synthesis based on the EM series and time-lapse FM series carrying different cell-specific markers reveals critical dynamic behaviors across scales of identifiable individual cells in the emergence of the primary neuropil, the nerve ring, as well as a major sensory organ, the amphid. Our study paves the way for systematic cross-modality data synthesis in *C. elegans* and demonstrates a powerful approach that may be applied broadly.

\*For correspondence:
irina.kolotueva@unil.ch (IK);
baoz@mskcc.org (ZB)

†These authors contributed equally to this work

**Competing interest:** The authors declare that no competing interests exist.

## Editor's evaluation

This paper very nicely tackles a methodological problem in aligning different types of datasets (EM and light microscopy) to image embryonic nervous system development in the nematode *C. elegans*. The paper is important from a methodological standpoint, and also provides novel insights into nervous system development that will be of general interest.

## Introduction

Imaging modalities, both evolving established methods like electron microscopy (EM) (*Helmstaedter, 2013*; *Chklovskii et al., 2010*) and fluorescence microscopy (FM) (*Sydor et al., 2015*; *Keller et al., 2015*) and emerging methods like spatial sequencing (*Takei et al., 2021*) and micro CT (*Ding et al., 2019*) provide unprecedented streams of data. This data has revolutionized our ability to observe the development, structure, and function of the nervous system. However, interpretation and synthesis of this data remain challenging. Combining multiple modalities typically provides complementary information yielding a more holistic view of structural and functional dynamics across spatial and temporal scales (*Ball et al., 2012*; *BRAIN Initiative Cell Census Network, 2021*). Despite this, systematic atlasing that explicitly combines modalities (*Vergara et al., 2021*; *Lalit et al., 2020*) is limited, in part due to the technical difficulties in cross-modality image alignment. Most automated alignment methods such as those used in brain and whole organism atlases focus on appearance-based approaches (*Vergara et al., 2021*; *Chiang et al., 2011*; *Oh et al., 2014*; *Chen et al., 2019*; *Kunst et al., 2019*; *Wang et al., 2020*) which align areas with similar appearances. However, in cross-modality applications there is

generally little shared appearance. Another class of alignment methods use anatomical landmarks, typically represented as points, and take known correspondences (*Hunnicutt et al., 2016*) or match points (*Lalit et al., 2020*; *Heckscher et al., 2014*; *Yemini et al., 2021*; *Toyoshima et al., 2020*; *Yu, 2021*; *Long et al., 2009*; *Chaudhary et al., 2021*; *Preibisch et al., 2010*) to compute an alignment. To account for complex variation of landmark positions (*Toyoshima et al., 2020*; *Insley and Shaham, 2018*), a variety of approaches are used to exploit relative position to improve matching accuracy (*Lalit et al., 2020*; *Yu, 2021*; *Long et al., 2009*; *Chaudhary et al., 2021*; *Preibisch et al., 2010*), but performance is limited with dense landmarks (typically 50–60% accuracy, at best). Efforts to build in additional information such as multi-color labeling to distinguish the landmarks, while technically difficult, can greatly improve matching performance (to 81–86%) (*Yemini et al., 2021*; *Chaudhary et al., 2021*).

Developmental data, which encompass dynamic anatomical changes over an extended time period, contain even more challenges. Different developmental processes show slight variation of developmental speed, creating heterochrony among anatomical structures. Between samples of comparable developmental stage, certain cell types and tissues/organs may or may not be present, causing a mismatch in the landmarks present. These potentially significant differences between samples complicate alignment and matching. More quantitatively, heterochrony in cell movement (from active migration or tissue shape change) may cause systematic positional variation that confounds approaches with naïve assumptions. Alignment using manually selected landmarks that are always present across samples partially bypasses these challenges (*Wong et al., 2015*). Efforts at alignment that largely ignore these discrepancies between individuals have also been attempted, with limited accuracy (<50%) (*Lalit et al., 2020*). However, generally, heterochrony and the associated challenges in developmental data are unaddressed.

Here, we address these limitations in a cross-modal study of *C. elegans* neural development. Specifically, we address two methodological difficulties, the challenge of annotating identities in developmental EM data, and that of collating the spatial detail of EM with the temporal detail of FM. We build an EM time series of *C. elegans* embryonic development and conduct cross-modality synthesis with time lapse FM to examine neural development with simultaneous high spatial and temporal resolution.

EM studies in *C. elegans* have made profound contributions to neuroscience including the first connectome *White et al., 1997*, sexual dimorphism (*Cook et al., 2019*), variability among individuals (*Brittin et al., 2021*; *Witvliet et al., 2021*), and principles of structural organization (*Cook et al., 2019*; *Brittin et al., 2021*; *Witvliet et al., 2021*; *Moyle et al., 2021*). However, these efforts are focused on the larval and adult nervous system, which leaves unanswered many questions about how the complex structure of the nervous system arises. Some EM work examines distinctive embryonic structures that can be readily identified such as the excretory system (*Soulavie et al., 2018*), amphid dendrite tip (*Low et al., 2019*), or the general appearance of the ring (*Rapti et al., 2017*). However, since neurons are typically identified in EM based on specialized adult morphologies it is challenging to identify them in the embryo during their emergence. It is primarily work using live FM that has examined the *C. elegans* nervous system during embryogenesis, providing insight into neurulation (*Shah et al., 2017b*; *Barnes et al., 2020*), organogenesis (*Low et al., 2019*; *Fan et al., 2019*), neuropil formation (*Moyle et al., 2021*; *Rapti et al., 2017*; *Kennerdell et al., 2009*; *Shah et al., 2017a*; *Sengupta et al., 2021*), synaptic specificity (*Berghoff, 2021*), as well as lineage differentiation and brain asymmetry (*Chuang et al., 2007*; *Cochella and Hobert, 2012*; *Masoudi et al., 2021*). Lineage tracing in FM establishes definitive identities for cells via ancestry even when they lack distinctive positions or morphologies (*Shah et al., 2017a*; *Bao et al., 2006*) and light sheet microscopy allows imaging to extend into embryonic motion (*Wu et al., 2013*). However, these efforts are limited by the lack of spatial resolution and availability of cell-specific markers needed to resolve individual cell shape. To fill in this gap, we generate an EM embryonic series, develop and apply a landmark-based co-optimization algorithm for cross-modality alignment in the presence of developmental variation. Furthermore, we synthesize the EM series and live FM series data representing over a dozen markers. The cross-modality synthesis reveals critical dynamic behaviors of identifiable individual cells across scales in the emergence of the primary neuropil, the nerve ring, as well as the sensory end of a major sensory organ, the amphid. Our study paves the way for systematic cross-modality data synthesis in *C. elegans* and demonstrates a powerful approach that may be

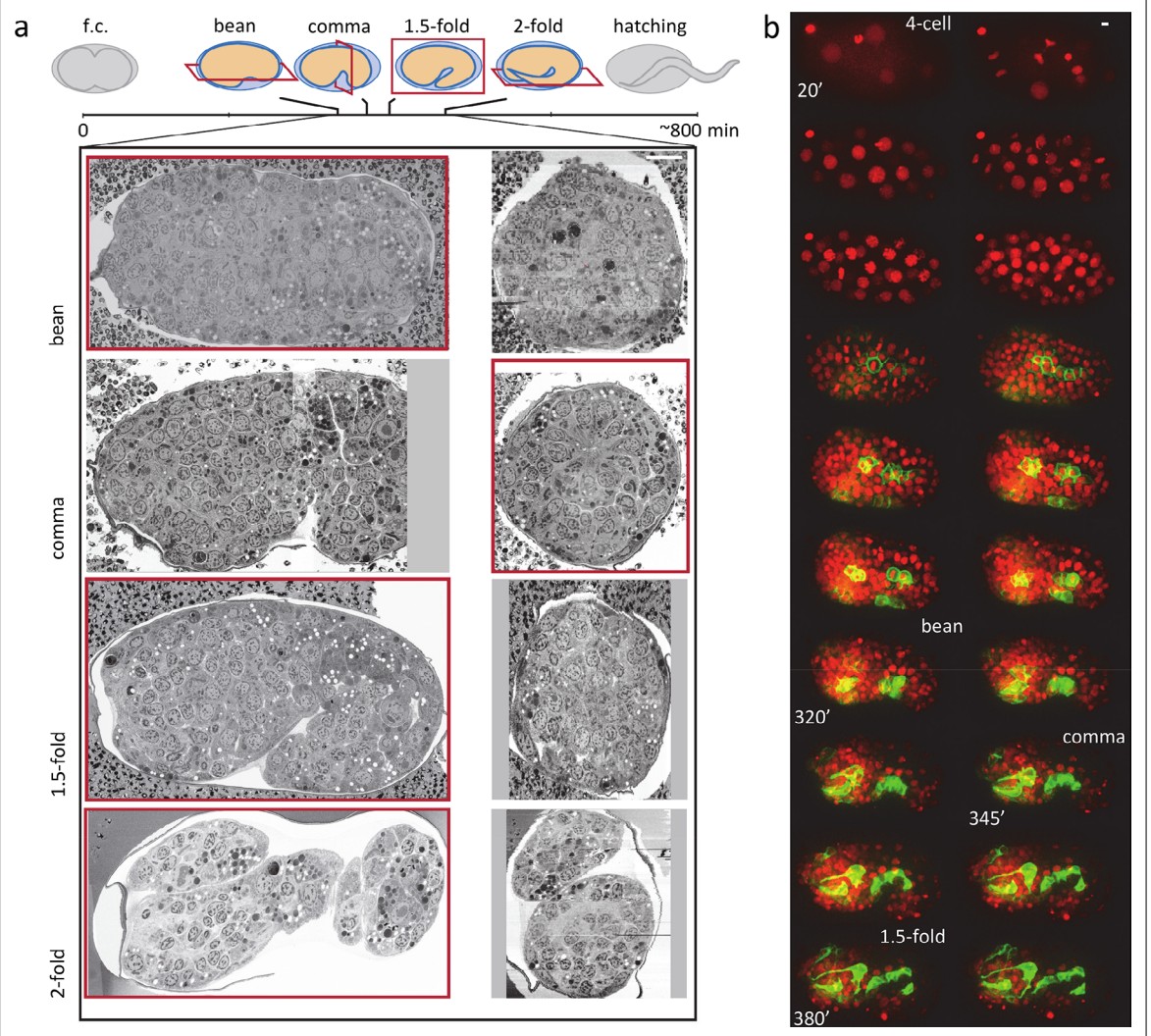

**Figure 1.** EM and FM time series of *C. elegans* embryogenesis with complementary spatial and temporal resolution. All scale bars indicate 5 µm. (**a**) Overview of EM data and their placement in the time course of *C. elegans* embryogenesis. Cartoons show body shape and the approximate physical sectioning plane orientation is shown in red. Orthogonal views of each dataset are shown below, with the view corresponding to physical sectioning outlined in red. f.c., first clevage or cell division. (**b**) An example FM series (see *Figure 1—video 1*) with a ubiquitous histone marker (red) and a *cnd-1* promoter driven membrane marker labeling a subset of neurons (green). Representative time points are shown as max projection. Time stamps are min post first cell division (pfc).

The online version of this article includes the following video and figure supplement(s) for figure 1:

**Figure supplement 1.** Overview of FM datasets.

**Figure 1—video 1.** FM embryo.

https://elifesciences.org/articles/77918/figures#fig1video1

applied broadly. We present the resulting annotated series as a resource for the community in an online format that facilitates additional community-based curation.

## Results

### Multi-modal developmental time series of *C. elegans* embryogenesis

We first image an EM time series of *C. elegans* embryogenesis with data at four classic stages, the bean, comma, 1.5-fold and 2-fold stages (*Sulston et al., 1983*; *Figure 1a*). The EM series spans a critical 150-min time window during which most tissues undergo major morphological changes: pharyngeal and intestinal lumens develop, distinctive morphologies develop in muscles and the excretory

system, the epidermis spreads and closes over the embryo, and the body length doubles. In the nervous system, most of the major nerve tracts emerge. The nerve ring is not yet present at the first time point, but densely populated at the end of this period. As part of the effort to test and demonstrate the feasibility of EM time series acquisition, the bean and comma stage embryos are imaged with Focused Ion Beam-Scanning Electron Microscopy (FIB-SEM) while the 1.5- and 2-fold stage embryos are imaged with Array Tomography (AT). The lateral resolution ranges from 8.5 to 19.4 nm, and z spacing 25–85 nm (see *Supplementary file 1a* and Methods). Entire embryos were imaged, containing 576–601 cells including visible apoptotic cells (*Supplementary file 1b*).

3D time-lapse FM is widely used to study *C. elegans* embryogenesis, providing rich information for cross-modality synthesis with EM. We focus on two-color FM images from the WormGUIDES project (*Moyle et al., 2021*; *Santella et al., 2015*), where a ubiquitously expressed histone/nuclear marker is used to label and track every cell and a sparsely expressed cell membrane marker reveals dynamic cell behaviors such as neurite outgrowth (*Figure 1b*, *Figure 1—figure supplement 1*; *Moyle et al., 2021*; *Barnes, 2020*). These image series typically begin at the four-cell stage and continue for approximately 8 hr until the 1.75-fold stage and the onset of embryonic movement, with a temporal resolution of 60 or 75 s. The nuclear labels are used to trace the entire cell lineage and provide the position and lineage identity of every cell at every time point (*Santella et al., 2014*; *Katzman et al., 2018*) (see *Figure 3—video 1*).

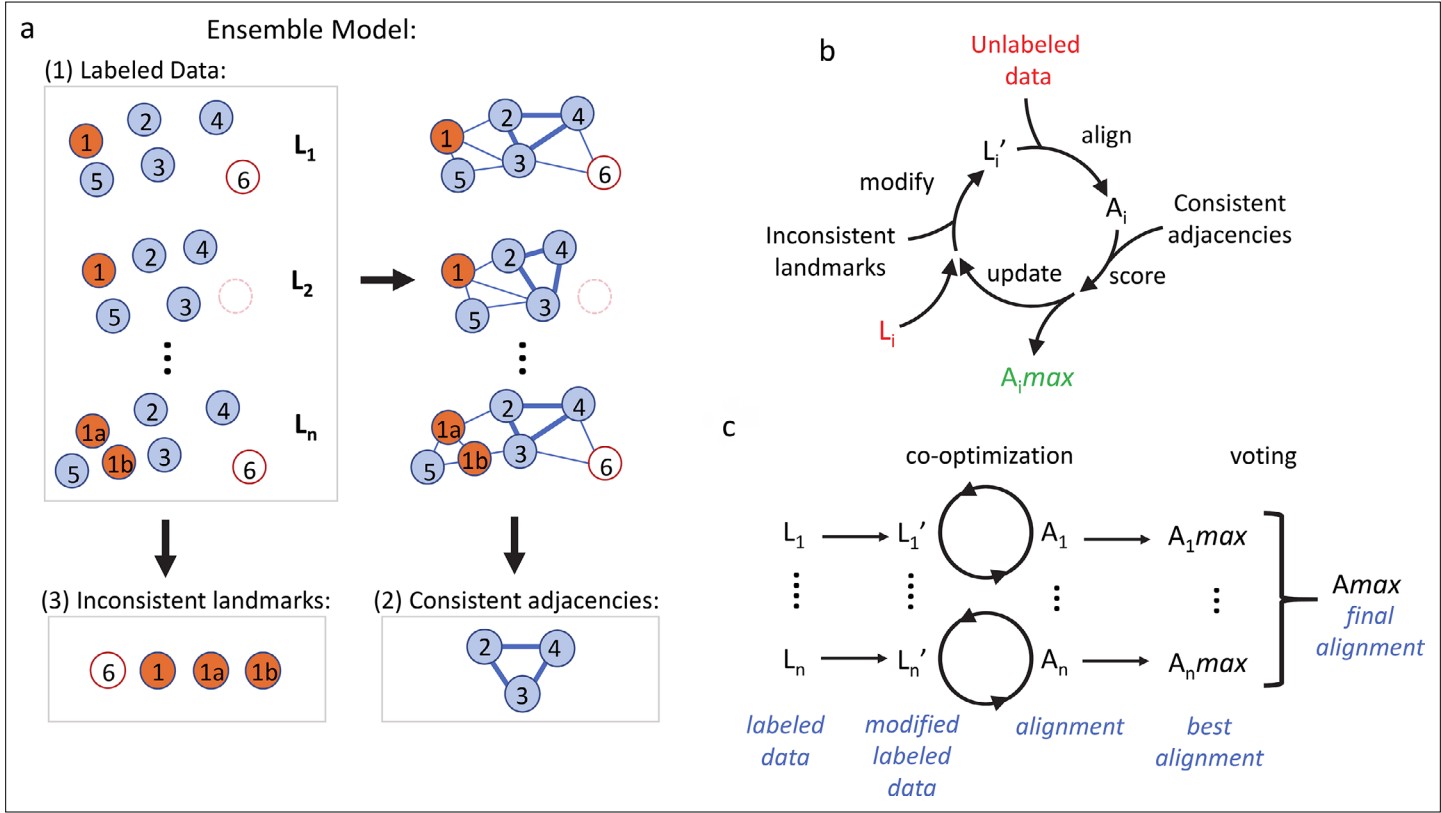

**Figure 2.** The co-optimization algorithm for cross-modality alignment of developmental data. (**a**) Components of the ensemble model. Circles represent landmarks. Numbers in each circle mark the corresponding landmarks across individual labeled data ($L_1$ to $L_n$). 1a and 1b denote a split of 1 (cell division or other separation of one landmark into two) and the empty circle indicates the disappearance of 6 (cell death or other transient landmark disappearance). Lines represent computed adjacencies, among which thick lines represent consistent adjacencies. (**b**) Co-optimization of landmark composition in a labeled dataset $L_i$ and alignment $A_i$ between $L_i$ and unlabeled data. $L_i$ is iteratively modified to $L_i'$, aligned, and scored to judge each modification. Modifications that improve score are accumulated to yield an optimized $L_i'$ and corresponding alignment $A_i max$. (**c**) The overall process of deriving correlated identities for an unlabeled dataset. Each labeled dataset $L_i$ is optimized to generate a best corresponding $A_i max$. Voting per landmark is then performed over the set of labels across $A_i max$ to achieve a final labeling Amax.

The online version of this article includes the following figure supplement(s) for figure 2:

**Figure supplement 1.** Method details.

In order to correlate a FM series to the EM series, we use the nuclear positions, which are also annotated for the EM series, as the common reference (i.e. a Rosetta stone) which allows us to identify cells in the EM. Given nuclei in both data sets we find the optimal 4D (3D+time) alignment between cells in the FM and EM series which conveys the lineage based identities to cells in the EM series. We then relate the dynamic cell behaviors as observed in the FM and EM series based on the correlated cell identities.

## The co-optimization algorithm for cross-modality alignment of developmental images

We present a landmark-based approach for cross-modality image alignment, which accounts for developmental variation and functions well with challenging dense landmarks. Alignment here takes the form of assigning landmark identities from a set of labeled data to novel unlabeled data, this involves both the computation of a geometric alignment proper and matching. The landmarks, which could be of any anatomical scale that originate from any imaging modality and segmentation method, are represented as spatial point clouds. As detailed below, our algorithm involves an ensemble model of example data to represent the anatomical structure of landmarks in the labeled data and a co-optimization approach for alignment.

The ensemble model contains three parts (*Figure 2a*) in order to capture the regularity and developmental variations of the anatomy: the set of labeled data; a neighbor graph representing consistent estimated adjacencies (see Materials and methods) observed between landmarks across individual labeled datasets; and the set of variable landmarks whose presence is inconsistent among labeled data. First, the ensemble, like instance-based learning approaches that use training data to directly model a distribution, can better preserve the complex positional variation observed in the labeled data. Second, the consistent adjacencies graph summarizes the structural consistency in three-dimensional spatial relationships. The adjacency graph captures relative positions, which makes it insensitive to body size variation, or shifting between organs and tissues due to posture, sample distortion or heterochrony in development, since the invariant positional relationships within a given organ/tissue are preserved in the graph. The approach is further motivated by biology. Physical adhesion between cells and tissues underlie anatomical regularity. Consistent adjacencies over multiple samples enrich for the bona fide regularity caused by some underlying physical driver and removes coincidental adjacencies. Third, and most importantly, the set of inconsistent landmarks provides a data-driven approach to handle discrepancies in landmarks between labeled and unlabeled data, which has been a major challenge and source of error in using alignment to establish identity via matching. Inconsistent landmarks are likely discrepancies, which can be used to deliberately modify a labeled dataset by adding or removing these landmarks to reduce discrepancies and improve the success of alignment and identity assignment.

The co-optimization algorithm uses the ensemble model to simultaneously optimize modifications to the set of landmarks in labeled data and the alignment between the modified labeled and the unlabeled data (*Figure 2b*). Adding or removing inconsistent landmarks in labeled data $L_i$ creates a modified $L_i'$ to be aligned with the unlabeled data. We score the resulting alignment by counting missing consistent adjacencies. This score is used as an objective function to drive optimization as we modify the set of inconsistent landmarks. The optimized set of landmarks ultimately yields an alignment with a maximal score, $A_i max$. We term this process by which we optimize to pick a set of modifications, scoring them by the alignment quality co-optimization. Finally, to utilize the multiple sets of labeled data in the ensemble model, the unlabeled data is independently aligned with each labeled dataset. From the alignment with each labeled dataset: $A_i max$, each landmark in the unlabeled data receives a predicted identity. A consensus $Amax$ is arrived at by a vote among the set of predicted identities for each landmark, where the most common identity is taken (*Figure 2c*). The co-optimization algorithm is implemented with a correspondence-free nonlinear warping method for pre-alignment (*Myronenko and Song, 2010*; *Figure 2—figure supplement 1a*), Linear Assignment Problem (LAP) matching (*Jonker and Volgenant, 1987*), data-driven modification, and greedy optimization of labeled data (see Materials and methods).

## Correlated cell identities convey a single-cell view of EM series

We apply the co-optimization algorithm to align the FM and EM series of *C. elegans* embryogenesis, using individual cells/nuclei as landmarks. We use the FM series, where cells were tracked and lineage

identities assigned, to generate the ensemble model for each developmental stage in the EM series and transfer cell identities to the EM (see Materials and methods). Due to the lack of lineaged FM data at the 2-fold stage, we align only the bean, comma and 1.5-fold stages in the EM series. Based on the timing of known developmental events in the FM series, the aligned EM data are timed as approximately 320, 345, and 380 min post first cell division (pfc), see Materials and methods for details. During this developmental window, 75 cell divisions and 76 deaths occur, while cell counts remain virtually constant (576-601) (*Figure 3a*). The ensemble FM model for each EM dataset (*Supplementary file 1c*) contains 39 labeled 3D FM images of embryos, each generating 5838–6170 pairwise adjacencies between landmarks, of which 439–504 (~8%) are consistent, as well as 27–34 inconsistent landmarks/cells (~5% of cells) (*Figure 3a*, *Figure 3—figure supplement 1a*, *Figure 3—figure supplement 2a*). While the scale of the ensemble models is fairly constant over the developmental stages, the consistent adjacency graph that guides alignment is dynamic (*Figure 3*, *Figure 3—figure supplement 1a*, *Figure 3—figure supplement 2*), only 146 adjacencies, about a third of consistent adjacencies, are observed at all three periods due to both inconsistent landmarks and embryo-wide cell movements during organ morphogenesis and body elongation (*Barnes et al., 2020*). We note our method uses only perfectly consistent landmarks but estimated adjacency captures more granular information about the consistency of cell positions, and has other potential applications in quantifying the consistency and variability in development (*Figure 3—figure supplement 2b, c*, *Supplementary file 1*).

The result is a single-cell level alignment between the FM and EM series, with a predicted identity for each cell in the EM series (*Figure 3b*, *Figure 3—figure supplement 1b*), which we refer to as the correlated EM series. We assess the accuracy of the correlated cell identities based on a set of manually curated cell identities across organs and tissues (*Figure 3—figure supplement 1c*, 148 to 242 per embryo). We consider accuracy at two levels: individual cells and tissues (*Figure 3c*). Single cell accuracy, which is the agreement between manual and predicted identity, varies between 71% and 78%, while accuracy of tissue identities is 79–95%, both with a trend for decreasing accuracy for the later developmental stages. Among head neurons, the single cell accuracy is 69–78%. There is no comparable embryonic work, but this is higher than previous reported efforts that attempted to align and name all head neurons in young adults based purely on cell positions (50–68%) (*Yemini et al., 2021*). As an additional validation we use our approach to predict the names for all cells from anonymized nuclear positions at each stage in three additional FM embryos (see Materials and methods) and show comparable success (*Supplementary file 1f*).

These results are deposited on webKnossos (*Boergens et al., 2017*) as a community resource containing the EM images, annotated nuclear positions and cell identities. Data is publicly accessible and can be navigated within a convenient web interface that allows users to search for cell identities by name or query a cells identity by clicking on its centroid in the image. With a free account users can fork the annotation and edit names or reconstruct neurons or other structures. Currently, 56–70% of the cells have identities that are manually curated (*Figure 3—figure supplement 1c*), including 25–40% as cell identities (see *Supplementary file 1e*) and 16–34% as tissue/organ identities, greatly reducing potential errors in identities. Annotation will be continually updated based on community feedback and our own ongoing efforts at neuronal reconstruction, which we hope will make these online data sets an increasingly useful community resource. Researchers requiring raw data access are encouraged to contact the authors.

The correlated cell identities convey a single-cell view of the EM data for developmental dynamics at different scales (*Figure 3d and e*). As an example of how developmental succession can be followed over time we highlight tube formation in the excretory system. The excretory pore, canal and duct can be seen stacked on top of each other at the bean and comma (*Figure 3d and e*) stages. At 1.5-fold a continuous tube is visible that connects the canal to the outside environment which has formed through concerted auto fusion of these cells (*Figure 3f and g*). These events are captured with a clarity comparable with EM imaging taken specifically to study these events (*Soulavie et al., 2018*; *Sundaram and Buechner, 2016*).

## Spatial and temporal organization of neuropil formation

Leveraging the co-optimization algorithm for alignment, we conduct cross-modality synthesis of diverse data sets documenting early NR development to elucidate the spatial and temporal organization

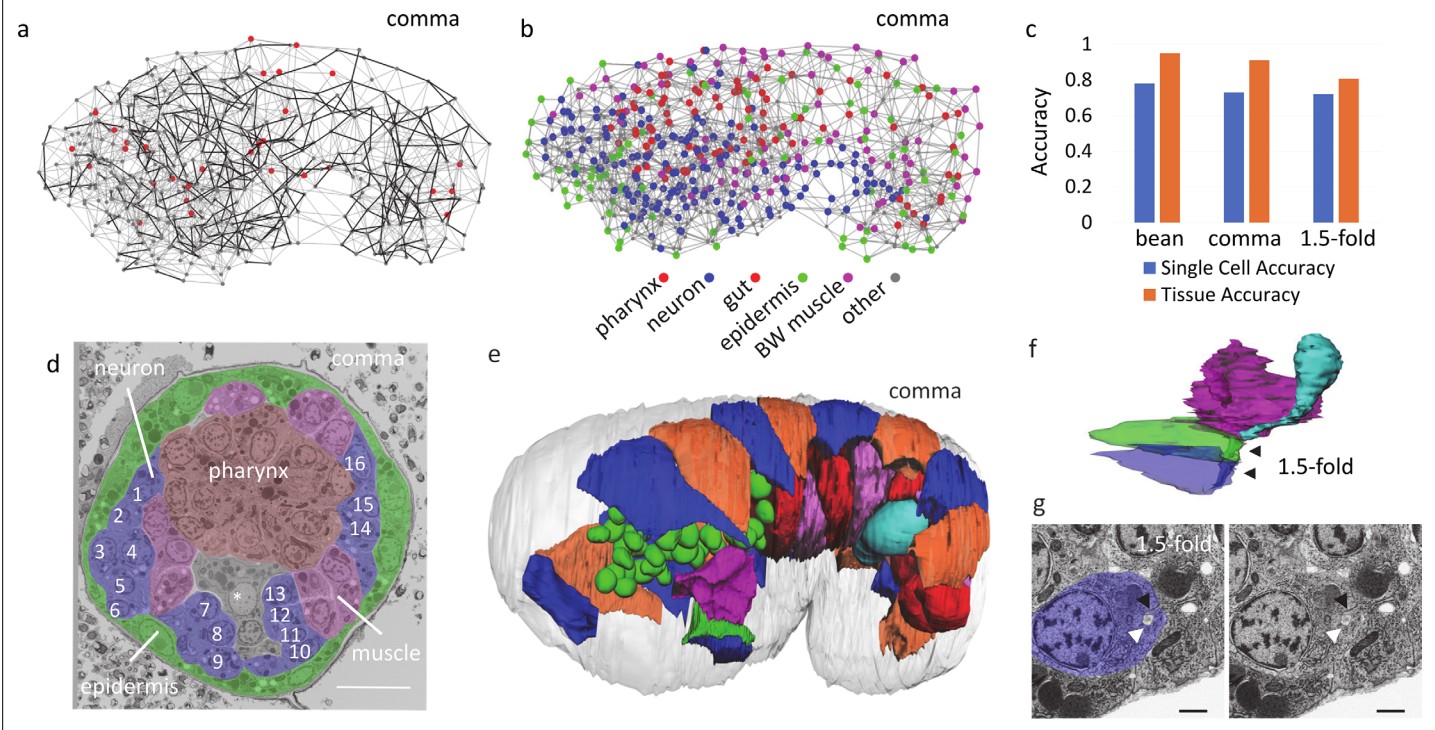

**Figure 3.** Cross-modality alignment conveys a single-cell view of EM data. (**a**) Illustration of the ensemble model for the comma stage. Landmarks (dots) and adjacencies (edges) in a single instance of labeled data. All adjacencies are shown with the consistent adjacencies highlighted in black. Inconsistent landmarks are larger and red. (**b**) A visualization of the comma stage EM data after alignment. Nucleus centroids are dots colored by major predicted tissue type with color key provided. Adjacencies are shown as gray lines. (**c**) Accuracy of predicted identities for each EM data at the single-cell and tissue levels. (**d**) Illustration of EM data annotated by identities. Tissue regions are shaded following the color scheme in b to highlight overall anatomical structure. The excretory canal cell is marked with a star with the nucleus of the excretory duct and part of the pore cell body visible below it. Individual neurons are numbered. Identities as follows are from alignment (-a) or manually confirmed (-c) 1 FLPR/AIZR parent-c, 2 ASER-c, 3 AVBR-c, 4 ASHR-c, 5 AWCR-c, 6 SIBDR-c, 7 AVKR-a, 8 AIYR-a, 9 SMBDR-c, 10 SMBVL-a, 11 AIML-a, 12 AVKL-a, 13 SMDDL-c, 14 FLPL/AIZL parent-a, 15 RMGL-a, 16 ALML/BDUL parent-c. Scale bar indicates 5 μm. (**e**) 3D reconstruction in comma stage EM data colored to maximize local contrast and showing cell body contours for seam cells (blue and orange), gut cells (alternating shades of red), germ line (cyan), and the excretory canal, duct and pore cells (red, green, and blue respectively) as well as nuclear contours for pharynx (green). (**f**) 3D reconstruction of the excretory system in the 1.5-fold stage EM data. Red, green, blue and cyan are the excretory canal, duct, pore and gland cell, respectively, as in e. Black arrows point to auto fusion in the duct and pore cell. (**g**) EM view of lumen (white arrow) and site of auto fusion (black arrow) in the excretory pore cell at 1.5-fold stage. Scale bar indicates 1 μm.

The online version of this article includes the following video, source data, and figure supplement(s) for figure 3:

**Figure supplement 1.** Cross-modality alignments.

**Figure supplement 2.** Consistency and change of adjacency over time.

**Figure supplement 1—source data 1.** EM annotation bean stage.

**Figure supplement 1—source data 2.** EM annotation comma stage.

**Figure supplement 1—source data 3.** EM annotation 1.5-fold stage.

**Figure supplement 2—source data 1.** All adjacency information bean stage.

**Figure supplement 2—source data 2.** All adjacency information comma stage.

**Figure supplement 2—source data 3.** All adjacency information 1.5-fold stage.

**Figure 3—video 1.** 3D rendering of embryo landmarks.

https://elifesciences.org/articles/77918/figures#fig3video1

underlying neuropil formation. The nerve ring (NR) is the primary neuropil in *C. elegans*. The adult NR consists of 181 neurites, where neurite topography and contacts constrain synapse formation and circuit function (*Brittin et al., 2021*; *Witvliet et al., 2021*; *Moyle et al., 2021*; *Durbin, 1987*). The NR has long been a prime model to study the general principles of brain structure and function. The NR emerges in the later half of embryogenesis (*Rapti et al., 2017*; *Altun et al., 2021*), but our understanding of the spatial and temporal organization of neurite outgrowths in this process is still

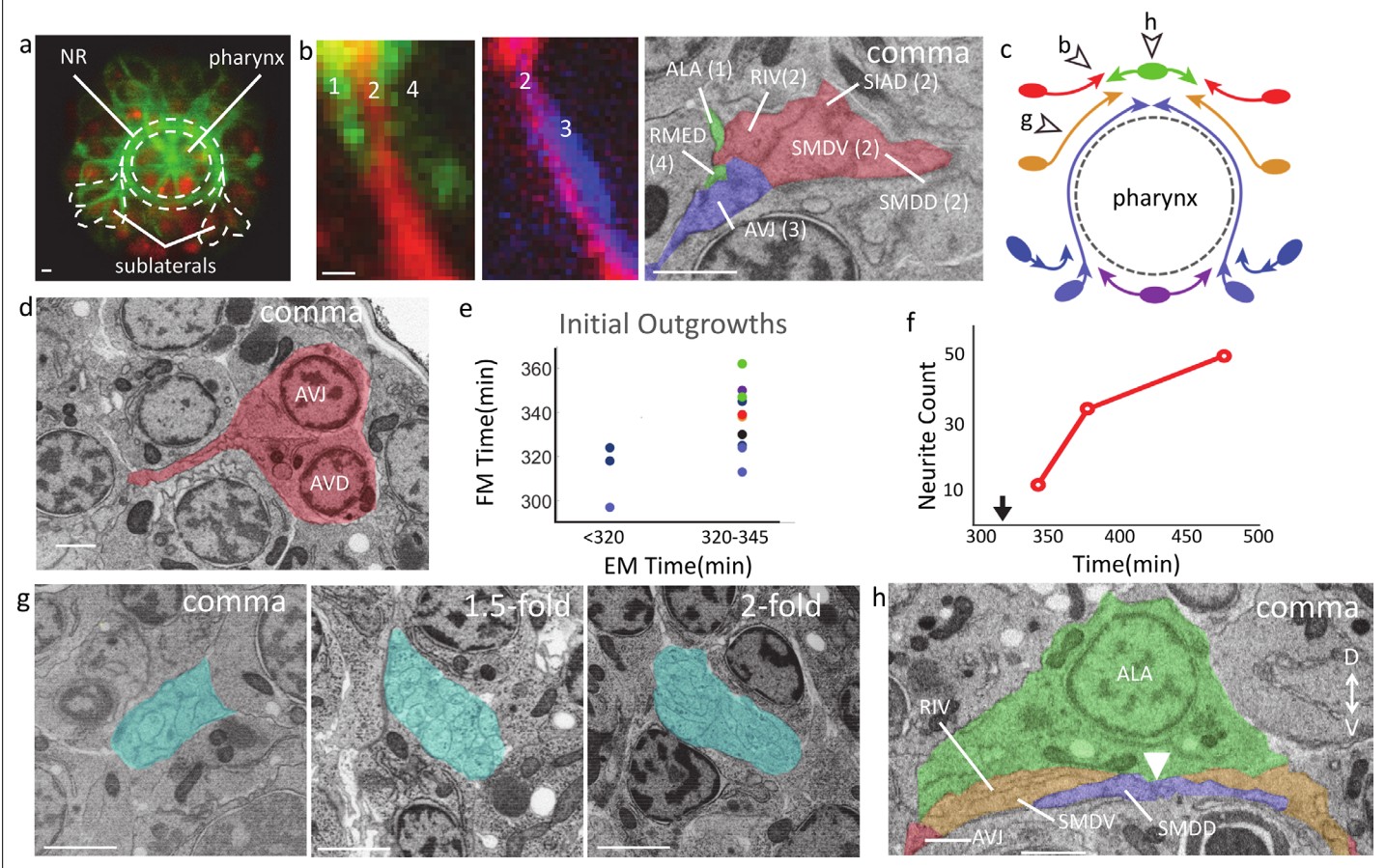

**Figure 4.** Spatial and temporal organization of neuropil formation. All scale bars 1 μm. (**a**) FM cross section of a comma stage embryo with a ubiquitous histone marker (red) and a broadly expressed cell membrane label (green) (*Barnes et al., 2020*). Anterior view. Dotted line encircles the NR and routing of sublateral neurons into the ring. (**b**) Comparison of resolution in resolving NR in FM and EM. Two images of two-color FM image of developing neurites (red, green, and blue are labeled by the promoters of *cnd-1*, *ceh-10*, and *zag-1*, respectively). Neurite bundles are numbered anterior to posterior. EM showing the corresponding neurites and their placement in the ring at the comma stage. Neurites are labeled with names and colors of the corresponding bundle in FM. (**c**) Schematic of the NR (anterior view) showing first outgrowths, entry points, and approximate extent of neurites at bean and comma stages. Ovals represent neuronal soma and arrowheads the leading edge. Two arrowheads are shown for the amphid (dark blue) and sublateral (light blue). The earlier arrowhead indicates initial outgrowth, and the latter line and arrow indicates the entry points and approximate outgrowth extent at the bean stage. Moving dorsal to ventral, dorsal outgrowth is green, supralateral red, lateral orange, and ventral purple. See *Figure 4—figure supplements 1 and 2* for corresponding EM. Open arrows and associated letters indicate the figure panels that show the location pointed to. (**d**) EM at comma stage showing the first outgrowth at the supralateral entry point with cells shaded red and named. (**e**) Comparison of initial outgrowth timing derived from FM and EM. Each dot represents a single neuron or small cluster of neurons. Dot color corresponds to membership in groups as pictured in panel c. Cells that do not clearly correspond to these major entry points are gray (See *Supplementary file 1d* for details). (**f**) Number of neurites visible in the cross sections of the ring in panel g. The location of this cross section is indicated by arrow b in panel c. Black arrow indicates bean stage. (**g**) Axial NR cross sections (neurites shaded cyan) near the lateral midline entry point (the location indicated by arrow g in panel c). (**h**) Precision in neurite outgrowth dynamics. A resliced view of the dorsal midline at comma stage shows the point left and right sublateral early neurites meet (white arrow). Colors match schematic in c.

The online version of this article includes the following figure supplement(s) for figure 4:

**Figure supplement 1.** Earliest neurite outgrowths.

**Figure supplement 2.** Early neurites at the comma stage.

poor. Efforts with FM revealed that the first neurites appear around the bean stage and a ring around the pharynx becomes visible by the comma stage (*Figure 4a*; *Moyle et al., 2021*; *Rapti et al., 2017*). FM with cell-specific markers and sparse labeling has led to critical insights: identification of pioneers (*Moyle et al., 2021*; *Rapti et al., 2017*; *Kennerdell et al., 2009*), the inside-out pattern of organization of outgrowth (*Moyle et al., 2021*), and even neurite relative placement through three color (*Yip and Heiman, 2018*) or combinations of two-color imaging (*Figure 4b*). However, lack of markers to

systematically label the entire nervous system and lack of spatial resolution hinder further insights. EM on the other hand provides a global view of not only every neurite but also the surrounding tissues with high spatial resolution (*Figure 4b*). We scan through EM data to identify neurites, and back trace them to soma in order to obtain identities (see Materials and methods). We also correlate EM with a set of FM series that encompasses 12 strains with cell specific markers that label neurons individually or in small, isolated groups (*Figure 1—figure supplement 1*, *Supplementary file 1d*; *Moyle et al., 2021*; *Barnes, 2020*).

Our analysis shows that the establishment of the NR is organized by eight entry points which include the dorsal midline, the ventral midline, and three left/right symmetric pairs in between, namely supralateral, lateral and sublateral (*Figure 4c*). The early time points in the correlated EM series reveal the first outgrowths that establish each entry point (*Figure 4c and d*, *Figure 4—figure supplement 1a, b, c*). Sublateral neurons, namely SIAD and SMDD, are the first neurons to initiate outgrowth into the ring, with their initial outgrowth captured at the bean stage (*Figure 4—figure supplement 1b*). This observation is consistent with these neurons serving as early pioneers for the NR (*Moyle et al., 2021*; *Rapti et al., 2017*; *Kennerdell et al., 2009*). In less than a half hour after this outgrowth (comma stage), all entry points have been populated with the corresponding early neurites. Moving from the dorsal midline to the ventral midline: at the dorsal midline, RMED and ALA have grown bilaterally (*Figure 4—figure supplement 2b*); supralateral neurons, namely AVD, AVH, and AVJ have entered and grown dorsally (*Figure 4d*, *Figure 4—figure supplement 2c*); lateral neurons, namely SMDV and RIV have also entered the ring almost reaching the dorsal midline (*Figure 4—figure supplement 2d*); and at the ventral midline, RIR, RMEV, and RIH have bilaterally advanced dorsally (*Figure 4—figure supplement 2g*). While the EM series provide a comprehensive and definitive set of entry points and first outgrowths due to its label-free nature, the FM series provide much finer temporal resolution of outgrowth dynamics. In terms of the timing of neurite initial outgrowth (*Figure 4e*, *Supplementary file 1d*), outgrowths observed as having occurred in EM by the bean stage span almost 30 min in the FM while outgrowths observed in EM by the comma stage span an hour as measured in FM. Most of the FM times are congruent with the upper bound of timing provided by EM. Further data would clarify if the occasional contradiction in timing is due to biological variability or technical issues in temporal alignment (see Materials and methods).

In addition to defining the first outgrowths and the initial outgrowth times, the cross-modality synthesis also reveals dynamics of subsequent neurite growth in the NR (*Figure 4f*). Assayed in a cross section near the lateral entry point (*Figure 4g*, arrow g in *Figure 4c*), the number of processes visible in the NR increase rapidly from 12 at the comma stage to 51 by the twofold stage. Outgrowth dynamics is not reducible to the timing of first outgrowth, subsequent extension also has striking temporal dynamics. For example, the first outgrowths into the amphid commissure, namely ASH, RIB, and AWC, show initial outgrowth at the bean stage, which make them one of the first neurons to grow out in the entire nervous system along with the sublateral neurons (*Altun et al., 2021*; *Figure 4—figure supplement 1a, c*). However, these neurites, and additional followers from the amphid neurons pause soon after their initial outgrowth (*Moyle et al., 2021*; *Barnes, 2020*) near the sublateral entry point (*Figure 4—figure supplement 2a, e* ) before they enter the NR (by the 1.5-fold stage).

Finally, the spatial resolution of the EM series highlights the precision of control in neurite outgrowth kinetics. For example, comparing the left- and right-side behavior of the first cells to grow into the NR, namely SIADL/R and SMDDL/R (*Figure 4—figure supplement 1b*, *Figure 4—figure supplement 2f*), suggests strong symmetry in behavior. Both sides have broken out at the bean stage, with similar sized outgrowths. The contralateral pairs then reach the midline simultaneously at the comma stage: their tips meet precisely beneath the midpoint of the ALA nucleus, which marks the dorsal midline (*Figure 4h*). This level of precision is not limited to the first sublateral outgrowths. Left and right counterparts among the lateral and supralateral first outgrowths have also reached nearly identical positions relative to the dorsal midline at the comma stage (*Figure 4h*). Quantification of the length of the most advanced sublateral neurite pairs SMDDL/R and SIADL/R shows left-right counterparts differ in both cases by less than 2 µm, a few percent of overall length. It is not clear how such level of precision is achieved, or if there is signaling to coordinate between the two sides at this stage though asymmetric sensitivity to guidance molecules has been shown in amphid axons (*Grossman et al., 2013*).

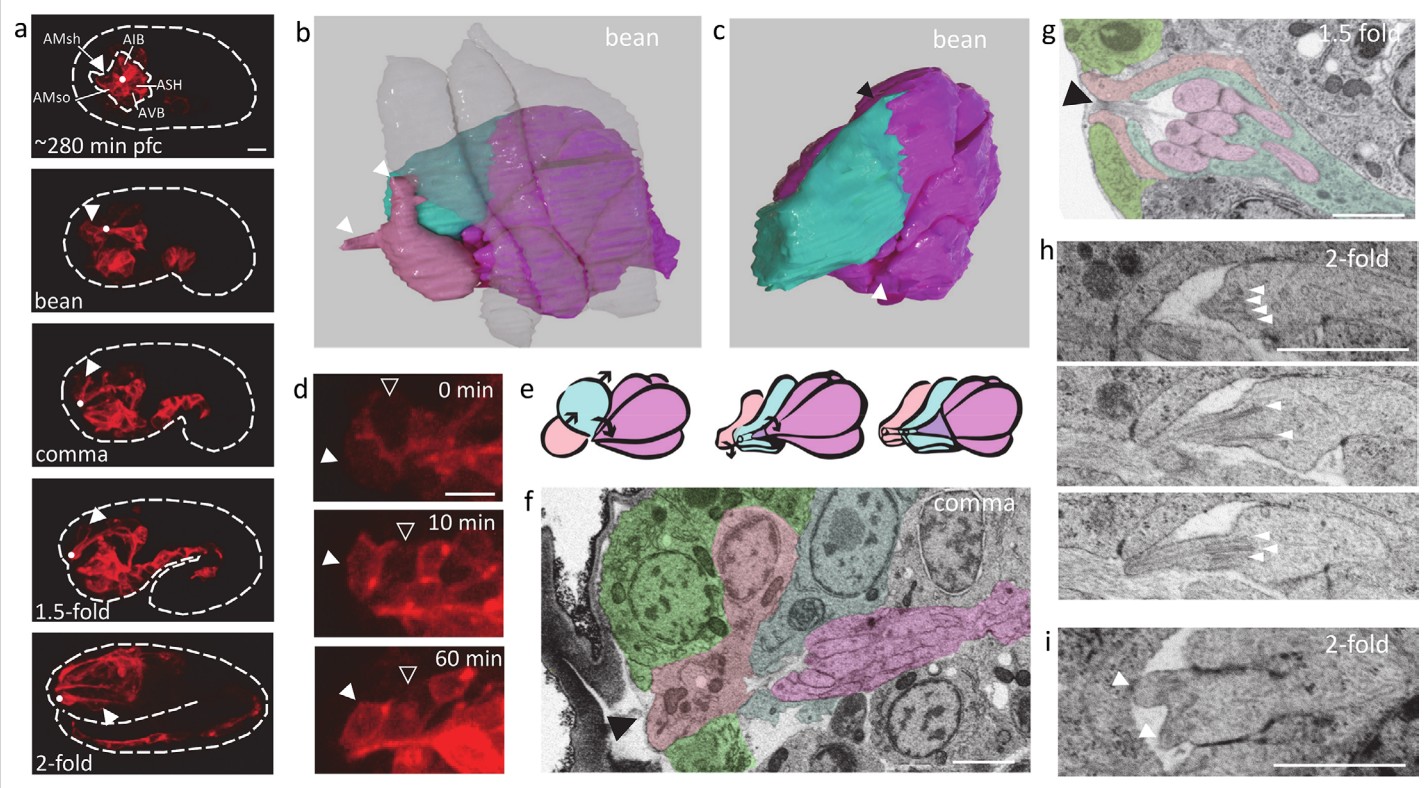

**Figure 5.** Cell-cell interactions and ultrastructure during amphid organogenesis. All scale bars 1 µm. (**a**) FM view of amphid dendrite development with a *cnd-1* promoter driven membrane label. Dashed line shows embryo contour. Dot indicates the location of dendrite tips, and arrow the socket and sheath cells. Dashed circle inside the embryo outlines the initial amphid rosette at ~280 min pfc. (**b**) 3D reconstruction of cells participating in the collective dendrite extension from bean stage EM. Epidermis: semi-transparent; amphid neurons: magenta; sheath cell: cyan; socket cell: pink. Arrows indicate protrusions on the socket cell. (**c**) The same reconstruction as in b with socket and epidermal cells removed. Black arrow highlights protrusion on the sheath cell. White arrow indicates focal point of amphid neuron protrusions. See also *Figure 5—figure supplement 1a-c*. (**d**) Partial max projection of FM image with a *cnd-1* promoter driven membrane label showing socket cell (filled arrow) and more weakly labeled sheath cell (open arrow) posterior migration. (**e**) Schematic of the time course of sensory opening formation. Arrows indicate motion of cells. Colors match b. (**f**) Resliced view of EM at comma stage highlighting socket cell (tinted rose) shape and relative position to epidermal cells (green), sheath cell (cyan), and dendrites (magenta). Black arrow indicates contact between the socket cell and the exterior of the embryo. (**g**) EM at the 1.5-fold stage showing the sensory opening. Colors follow f. Arrow indicates the opening to the exterior of the embryo. (**h**) EM at twofold stage showing three sections of a dendrite tip. White arrows mark nine individual microtubule structures. (**i**) EM at twofold stage showing a bifurcated dendrite tip, each tip marked with an arrow.

The online version of this article includes the following video and figure supplement(s) for figure 5:

**Figure supplement 1.** Amphid organogenesis.

**Figure 5—video 1.** Socket, sheath cell migration.

https://elifesciences.org/articles/77918/figures#fig5video1

## Cell-cell interactions and ultrastructure during amphid organogenesis

The amphid is a major sensory organ of *C. elegans* consisting of 12 neurons and two glial cells (the socket and sheath cells). The adult sensory end is a bundle of ciliated dendrite endings with drastically different morphologies surrounded by a tube-shaped glial channel. The glial cell extensions are in turn embedded between epidermal cells to create a sensory structure that is open to the environment (*Altun et al., 2021*; *Ward et al., 1975*; *Doroquez et al., 2014*). The dendrites grow via a distinctive mechanism, a collective retrograde extension from an initial rosette structure towed by the migrating epidermis (*Low et al., 2019*; *Fan et al., 2019*; *Heiman and Shaham, 2009*; *Kunz et al., 2021*). Formation of the embedded opening and elaboration of dendrite morphology occur in embryogenesis (*Doroquez et al., 2014*; *Oikonomou et al., 2011*; *Nechipurenko et al., 2017*). Molecular components necessary for correct glial (*Oikonomou et al., 2011*; *Perens and Shaham, 2005*; *Oikonomou et al., 2012*) and cilium (*Bacaj et al., 2008*) formation have been identified. However, the detailed steps in the embryonic assembly of this structure are less understood.

The correlated EM series confirms the dynamic cell behaviors revealed by FM during dendrite extension (*Figure 5a*; *Fan et al., 2019*) and adds structural details. For example, 3D reconstruction at the bean stage shows small protrusions from amphid neurons clustered as in the initial rosette and lodged in a dent at the anterior/leading edge of the epidermis (*Figure 5b and c*, *Figure 5—figure supplement 1a, b, c*). Notably, the socket cell, though engaged in the rosette earlier does not appear engaged in this focal location at the bean stage and only touches a handful of amphid neurons (*Figure 5b*, *Figure 5—figure supplement 1a, b*).

More importantly, our cross-modality synthesis reveals critical details of cell behaviors and interactions in creating the multi-layered structure of the open sensory end (*Figure 5e*). The sheath cell starts on the anterior side of the amphid rosette (*Figure 5a*). Subsequently, the sheath cell body moves dorsal-posteriorly (*Figure 5d*). Lamellipodia-like protrusions from the sheath cell during this process are visible at the bean stage (*Figure 5c*). Meanwhile, the anterior portion of the sheath cell wraps itself around the collected mass of the dendrite tips, which is evident by the comma stage (*Figure 5f*). The socket cell exhibits more complex behaviors. One set of behaviors follows a similar course as the sheath cell. The cell body migrates posteriorly (*Figure 5d*), creating transient posterior protrusions (*Figure 5—figure supplement 1d*), while the anterior end remains close to the dendrite tips. Subsequently, after the sheath cell has wrapped the dendrites, the socket cell starts to wrap over the sheath cell (*Figure 5e and f*). The wrapping is complete by the 1.5-fold stage (*Figure 5g*; *Low et al., 2019*; *Oikonomou et al., 2011*). Distinctively, the socket cell must also interface with the epidermis to create the open sensory end (*Figure 5g*). Noteworthy details hint at the dynamic role of the socket cell in the process. During dendrite extension, the socket cell makes elongated protrusions, one of which appears to reach out and contact the advancing epidermal cell (*Figure 5b*). Later, the anterior end of the socket cell is juxtaposed between epidermal cells and exposed to the outside of the embryo like a plug (*Figure 5f*), which may play a role in creating the opening to the environment in the sensory end. The socket cell is fully surrounded by the epidermis by the 1.5-fold stage (*Figure 5g*, *Figure 5—figure supplement 1e*). Overall, the dynamic details from the cross-modality synthesis suggest that the strategy of amphid sensory end assembly is successive wrapping of interior structures with additional layers.

Furthermore, the series illustrates the elaboration of distinctive ultrastructure. Centrioles move to the dendrite tips between the bean and comma stages (*Figure 5—figure supplement 1f*; *Li et al., 2017*). By the twofold stage, centrioles have disappeared, presumably degraded or remodeled (*Nechipurenko et al., 2017*; *Li et al., 2017*), and microtubule bundles (9 bundles per dendrite) are visible growing into the short proto-dendrite tips (*Figure 5h*). Their appearance is essentially the same, aside from length, as adult structures (*Doroquez et al., 2014*). Meanwhile, the dendrite tips begin to elaborate their complex morphology. The shape of the dendrite tips remains simple up to the 1.5-fold stage. By the twofold stage, some diversity in length and shape is visible. Notably, two dendrites, namely ADF and ADL, show short bifurcated tips, each with its own set of microtubules branching from a central location (*Doroquez et al., 2014*; *Figure 5i*). Dendrites tips are not yet individually embedded within the sheath cell by the twofold stage.

## Discussion

Our study demonstrates a powerful co-optimization algorithm to align and assign identity to landmarks. Because each cell in *C. elegans* is unique, with well-characterized molecular and morphological traits, it is often desirable to identify individual cells. For postembryonic cell identification, people combine cell position with the characteristic cell/neurite morphology to make a positive identification. In the embryo, before such morphology becomes prominent, cell identification has been challenging. In his seminal work, Sulston suggested that cells in the late embryo can be identified by careful examination of relative positions within small neighborhoods (*Sulston et al., 1983*), but he remained the only person who could do so systematically. Instead, researchers use live imaging-based lineage tracing for reliable cell identification (*Bao et al., 2006*; *Schnabel et al., 1997*). By paying attention to relative position through the adjacency graph, our algorithm essentially follows Sulston's suggestion, and achieves unprecedented accuracy in cell identification. Additional information can be readily added to the adjacency graph to better score matched landmarks and improve the alignment, such as size, appearance and other features of the landmarks when available. Furthermore, while our study is focused on the wild type, one could use live FM to

build the appropriate ensemble models to analyze mutants. It should also be possible to adapt our methods to partially imaged embryos. Thus, by eliminating the bottleneck in data acquisition as well as data interpretation, we envision a broad use of EM and cross-modality synthesis of *C. elegans*. As a proof of concept, we present the image data and identity annotation of our correlated WT EM series as an accessible public resource. Existing annotation provides a navigational aid and the seed of a community effort that we hope will fully validate annotation of the series. Extending our method to mutants is conceptually similar, requiring positional information from several lineaged FM data sets and an EM time series. This data could be efficiently assembled, at least in the case of highly penetrant mutants.

More broadly, the co-optimization algorithm provides a general solution for cross-modality and cross-scale alignment. Representation of landmarks as a point cloud is modality- and scale-free. In our study, we use individual cells/nuclei as landmarks due to the desired single-cell resolution for *C. elegans* biology, but the point cloud can easily represent a mixture of landmarks at different scales, enabling ambitious future work. With advances in deep learning and image analysis, landmarks can be extracted from complex organs such as the mammalian brain by learned region detectors (*Chen et al., 2019*) with increasing reliability. While the mammalian brain is orders of magnitude larger in terms of cells, it is worth noting that in state of the art atlases it is currently annotated with hundreds of regions (*Wang et al., 2020*), which makes alignment at this level of detail a similar scale problem to ours. We postulate that at an optimal level of landmark density consistent adjacencies provide an effective implicit model of anatomy that can maximize available prior information about structure and constrain accurate automated alignment. The data-driven modification in the co-optimization algorithm further facilitates such a vision for future automated alignment and annotation of complex brain images. It readily handles not only biological variability in landmarks (developmental or between individuals) but also technical variability such as errors in landmark extraction or systematic differences between modalities. This model only becomes more powerful as data accumulates. Ultimately, we imagine a multi-modal fusion process built on our approach will be capable of systematically collating information from multiple modalities and discovering consistent landmarks, likely at the level of organs and anatomical structures rather than single cells. Our approach provides spatial alignment robust to landmark variability. Aligning between individuals and over time, and incorporating molecular information to allow segmentation of aligned landmarks into spatio-temporal regions would automate the decision making involved in defining the regions of a digital atlas (*Wang et al., 2020*) making a full spatio-temporal atlas of developing anatomy possible.

Last but not least, or experience provides broader insights on the usefulness of various forms of whole embryo EM in *C. elegans* and beyond. Firstly, our post-hoc correlation of EM and FM data provides a useful alternative to true correlative EM a powerful, but complex to implement, approach for targeted EM based on the expression pattern of fluorescence markers (*de Boer et al., 2015*). The identity of landmarks obtained from our post-hoc correlation can be used to select relevant targets for re-imaging with AT at higher resolution, similar to the use of correlative methods in identifying regions for EM imaging. Dense correspondences could also in principle be used to visualize correlated FM data superimposed on EM. Secondly, our efforts provide broad insights on the relative ease and usefulness of AT and FIB-SEM in the *C. elegans* embryo. FIB-SEM excels in axial resolution and avoids alignment artifacts, making it easier to follow small neurites between planes. This is however balanced by greater difficulty in achieving good results due to exacting preparation requirements and slow acquisition of a limited area. As the embryo is close to the maximal size of the imaging area there is danger that only late in acquisition will it become apparent that parts of the embryo are found to be out of the ROI captured. In addition, it can be difficult to precisely stage the embedded embryos until time has been expended on imaging them. In contrast, AT is more artifact prone, but a more flexible and forgiving process. With AT it is possible to section first, then inspect sections and select appropriate samples for imaging. In our application, this is particularly useful for selecting appropriately staged embryos from sections containing multiple embryos before spending time on imaging. AT acquisition is less automated and post-acquisition internal stack alignment can be challenging and require extensive manual effort. However, largely because of its reduced risk of acquisition failure, we have found AT to be more reliable in yielding useful data.

## Materials and methods
### EM imaging
### High-pressure freezing and freeze-substitution

A generous amount of *E. coli* was placed into 0.5 μm carriers filled with 20% BSA solution. Embryos from N2 worms were collected using a platinum pick from the seeded plates and settled in the carrier. The carriers were covered with a lid and frozen in a Wohlwend HPF Compact 02 (Wohlwend GmbH, Sennwald, Switzerland).

The freeze-substitution step was done in the freeze-substitution unit (AFS2; Leica Mikrosysteme GmbH, Vienna, Austria) following the previously described procedure (*Kolotuev et al., 2010*; *Lee et al., 2018*; *Kolotuev et al., 2012*). Briefly, frozen samples were transferred under liquid nitrogen to the freeze-substitution mix of 2% (w/v) osmium tetroxide, 0.1% (w/v), uranyl acetate 2% (v/v) H2O in acetone pre-cooled to –120 °C. Samples were heated to –90 °C over a period of 2 hr and left at this temperature for 2 hr. Over 24 hr, the temperature was raised to –30 °C. Samples were left in this temperature for 2 hr. After they were washed once with acetone and three times with ethanol, the temperature was raised to +20 °C. The solvent was replaced with the mix of ethanol and Epon resin mix (Embed 812 resin EMS, 1420), infiltrating the samples with increasing concentrations 30%, 50%, 75% for three hours each exchange at ambient temperature. Three incubations and exchanges of 100% EPON were done over 12 hr. Samples were flat embedded using the two-step procedure (*Kolotuev et al., 2012*) and polymerized for 48 hr at 60 °C.

### Block trimming

For accurate sectioning, it is important to orient samples correctly. Flat blocks facilitate recognition of a ROI and help in trimming. Delimited areas surrounding the sample were trimmed using a 90 diamond trim tool (Diatome, Biel, Switzerland). Care was taken to approach the sample as closely as possible and keep the top and bottom sides parallel to one another and the knife's surface (*Kolotuev, 2014*). A buffer area of empty resin at the leading edge of the trimmed sample was left to allow space for alignment maneuvers during the subsequent steps.

### Focused ion beam/scanning electron microscopy (FIB-SEM)

For FIB experiments, samples were oriented transversely or longitudinally already at the stage of trimming. Sample blocks with targeted trimmed rectangles ~100×200 μm were glued on aluminum SEM specimen stubs and coated with a 30 nm layer of platinum. Stubs were introduced into FIB-SEM (Helios Nanolab 650, Thermo Fischer Scientific) and further calculations were done inside the microscope chamber to target the ROI precisely to the embryo of interest. FIB/SEM acquisitions were performed using FEI Slice and View software. The FIB milling was done at 30 kV acceleration voltage, 790 pA current. Images were acquired with the in column detector (ICD, comma stage embryo) and mirror detector (MD, bean stage embryo) in immersion mode using an electron beam of 2 kV, 800 pA and 10 μs of dwell time.

### Array tomography (AT)

For AT experiments, the 'blind' sectioning strategy does not require precise sample orientation. Several samples can be sectioned at once, and final assessment of age is done after sectioning.

Planar arrays of the section were generated using the ATS knife (Diatome, Biel, Switzerland) mounted on a Leica UC7 ultramicrotome (Leica Microsystems; Vienna, Austria). Rectangles of silicon wafer approximately 2 × 4 cm (Ted Pella, No 16015) were cut from the round plate glow discharged, cleaned, and placed into the knife boat. 'Ribbons' of ~300 consecutive sections, 70–100 nm thick, were subdivided into shorter segments and then aligned side by side on support (*Kolotuev, 2014*). After the sections transfer and water drains, the wafer was let dry slowly at the ambient temperature and subsequently incubated for 30 min at 60 C.

The wafers were transferred to the Helios 650 FIB-SEM (Thermo Fischer Scientific). The strategy of positional correlation to select the embryos of the desired stage was used. To find the anchor sections that are worthy of being analyzed in sequence, we chose the strategy of "leaping" between centrally located sections across the short horizontally aligned 'ribbons' to screen for the

ROI. Acquisition parameters were similar to those used in FIB protocols. Images were acquired with mirror detector (MD) in immersion mode using an electron beam of 2 kV, 800 pA and dwell time of 5–10 μs.

## EM image analysis and rendering

For alignment of slices within an AT EM data set, we used IMOD. Automatic IMOD registration was performed. On top of this manual adjustment was used only to correct occasional qualitatively misaligned slices. 3D tracing of sections was done with the 3dmod module of IMOD (*Kremer et al., 1996*) or TRACKEM (*Cardona et al., 2012*) with 3D rendering done in Daz3D *Daz Studio, 2005*.

### 3D, time-lapse FM datasets

As previously described (*Moyle et al., 2021*) time lapse images of fluorescently labeled *C. elegans* embryos were collected with spinning disc or diSPIM microscopy (*Duncan et al., 2019*), and lineaged with StarryNite (*Bao et al., 2006*; *Santella et al., 2014*; *Boyle et al., 2006*) to assign cell identities. The annotated nuclear centers serve as landmarks for ensemble model creation and series correlation.

### Cross-modality alignment

## Pre-alignment

Pre-alignment between labeled and unlabeled data allows the use of absolute spatial position and is performed in two steps.

First, an approximate affine least squares or Thin Plate Spline (TPS) (*Zeller and Schempp, 1976*) alignment (the latter used for the 1.5-fold embryo) is performed based on a small number of manually identified landmarks. The landmarks are identified in each EM dataset based on their characteristic positions and cell shapes. Cells from different regions of the embryo are chosen to highlight the body axes (*Supplementary file 1g*). The landmark set was evaluated visually per EM data set to ensure the point clouds of labeled and unlabeled data fully overlapped when aligned.

Second, the approximate alignment is refined using a coherent non-linear method GMM-CPD (*Myronenko and Song, 2010*). This method creates a regularized local warping function to align two point clouds by iteratively warping and matching points under an assumption that nearby points move similarly, which provides a more granular adjustment for non-linear shifts in body proportion and relative positioning of organ structures (*Figure 2—figure supplement 1a*).

## Temporal staging

A coarse temporal staging is used to select the appropriate time window in the FM data from which to create the ensemble model. Stage is first visually estimated based on matching body axis elongation in the FM and EM datasets. At the comma and 1.5-fold stages this provides a very strong timing estimate. At the bean stage elongation is a weaker cue and we take advantage of the readily identifiable anterior gut cell Ealaa which undergoing mitosis in our bean stage EM embryo to refine our initial estimate. Isolated distinctive divisions are scattered throughout late embryogenesis making this a generally useful strategy. We then visualize the quality of spatio-temporal alignment by applying the pre-alignment as described above and plotting the EM aligned to a randomly selected FP data set. A visualization of manual correspondences under this pre-alignment is then used to adjust the temporal staging if alignment results appear to contain a systematic AP shift related to the elongating body axis.

We note that quantitative constraint violation scores from the alignment method described below can be used for more precise temporal alignment (*Figure 2—figure supplement 1b*) which would be useful in implementing fully automated methods of spatio-temporal alignment. A wide and smooth minima around the manual temporal alignment demonstrates that a minimization of constraint violations over different possible temporal alignments would provide a similar temporal alignment to our manual assessment. The absolute minima being offset by a few minutes from our manually assessed staging suggests the approach might even improve on manual time estimate. However, such a precise temporal alignment is not strictly needed for the purpose of choosing a relatively wide time window to create the ensemble model.

## Ensemble model creation

For each EM stage an ensemble model encompassing adjacency, position and inconsistent landmark information is created to be used in alignment. Adjacency and positional information is collected within a local (13 min) window of time in each of the three FM embryos that contribute to the labeled data ensemble. Temporal window size was optimized to maximize identity prediction performance using one of the three labeled data embryos as a test case and the other two as labeled data. Adjacencies are predicted from landmark positions via a Gabriel graph (*Gabriel and Sokal, 1969*) for all landmarks in a labeled dataset. Gabriel graph edges represent a subset of Voronoi diagram faces, a common computational approximation of cell membranes, that match an additional geometric definition of stability. Consistent adjacencies are those present in all datasets within the ensemble. A list of inconsistent landmarks is compiled from all cell deaths and cell divisions that occur within the local temporal window.

## Modification and co-optimization

We use our ensemble model to guide modifications of inconsistent landmarks in labeled data optimizing the exact set of labeled data landmarks along with alignment. This optimization uses missing consistent adjacencies in the unlabeled data, (measured on the predicted labeling of the unlabeled data which is created by the alignment) as an objective function in a process we term co-optimization.

Each possible modification to the labeled data represented by the list of inconsistent landmarks is tested. The hypothesis that a programmed cell death has already disappeared in the unlabeled sample is considered by removing that cell from the labeled data. For divisions the hypothesis that a recently divided cell in the labeled data has not divided in the unlabeled sample is considered by merging the two sibling cells into one parent cell at their midpoint. The hypothesis that a cell in the labeled data that will soon divide has already done so in the unlabeled dataset is entertained by splitting the parent cell into two cells sharing the same location. Note that while it would be possible to use prior information about the expected division axis to create more accurate daughter positions, this seems over complicated as the main goal is to prevent cascading errors due to qualitative discrepancies in the size of the set of landmarks present.

A simple, greedy, gradient descent process optimizes the labeled data. The optimization first considers the (limited in *C. elegans*) possible divisions and then deaths and accumulates changes that improve total adjacency constraint violations. For each of these hypotheses a match of the unlabeled dataset against the modified labeled data ($L_i'$) is made and adjacency constraint violations are counted under the corresponding alignment ($A_i$). A proposed change is kept if it results in a significant (in our case $\geq 3$) reduction in adjacency violations.

## Linear assignment and scoring via consistent adjacencies

For a given pre-aligned unlabeled dataset and modified labeled dataset, matching is framed as a linear assignment problem (LAP) minimizing the total distance between pairs of points. Missing consistent adjacencies are then used to score these matching results to evaluate modifications and to identify and prohibit incorrect matches.

Given an alignment between a modified version of the labeled data and unlabeled data that alignment can be evaluated by computing the Gabriel graph for the unlabeled data and, using the predicted identities under the alignment, counting missing consistent adjacencies. The total number of missing consistent adjacencies for the alignment is used to score the modification process described above.

In addition, a high number of missing consistent adjacencies for an individual landmark's predicted identity (summed over all expected adjacencies of that landmark only) indicates it is likely wrong (note this can be assessed only given the context of the global alignment, this cannot be assessed independently for a single landmark match). These individual match scores are used to identify and attempt to correct incorrect matches in an iterative process. The individual identity assignment with the worst score, (i.e. the most missing consistent adjacencies) if any exists, is made impossible (set to infinite cost) and LAP matching is repeated. This process is repeated for N iterations, and at the end of those iterations the global alignment with the lowest total violations is selected. Iterative error correction (200 steps) is performed after each modification, before judging its quality. Once a labeled dataset is fully co-optimized an additional 1400 iterations are performed and the best alignment and matching for this is $A_i max$.

## Voting and reiteration

Voting picks an identity for each landmark from the set of predicted identities corresponding to an alignment of the unlabeled data against each labeled data set. The most common name out of these independent answers for each landmark in the unlabeled data is taken as its identity. This procedure can result in multiple cells in the unlabeled data being assigned the same identity. Although it would be possible to force a unique set of identities by using an additional linear assignment between landmarks and available identities scored by total votes for each identity per landmark, we do not perform this step as we found this marginally reduces accuracy (data not shown).

Highly reliable landmark names can be used to reinitialize the alignment more accurately improving matching results. Landmarks in the sample where all or nearly all assignments agree are most reliable, (see *Figure 2—figure supplement 1c*). Taking landmarks where the winning identity has a 75% supermajority in place of the TGMM automated nonlinear alignment we compute a new landmark based TPS alignment based only on these 'known good' matches. These reliable automatic matches are greater in number than initial manual landmarks and can potentially provide a better nonlinear prealignment than TGMM. We then repeat the exact same co-optimization based matching for only the subset of uncertain cases, (the certain cases are reinserted with their fixed identities when evaluating adjacency constraints). All reported results include one round of iterative re-alignment.

Understanding of the overall structure of these steps is aided by pseudo code illustrating how these steps fit together:

Pseudo-code of the algorithm:

1. Alignment per labeled data: For each set of labeled data $L_{i=1..n}$
    • Two-step spatial pre-alignment between labeled and unlabeled data
        • Pre-Align via manually ID'd landmarks in $L_i$ and unlabeled data
        • Correspondence-less GMMCPD nonlinear pre-alignment
    • Modification and co-optimization of inconsistent landmarks:
        • $L_i$ max $=L_i$; $A_i$max $=$ LAP_with_iteration($L_i$ max, unlabeled data)
        • For each inconsistent landmark present in $L_i$
            • Make modification to $L_i$ max (adding or removing landmark) creating $L_i$ max'
            • $A_i =$ LAP_with_iteration($L_i$ max', unlabeled data)
            • Score($A_i$)=missing consistent adjacencies in $A_i$
            • If Score($A_i$)<Score($A_i$max)
                • $L_i$ max $= L_i$ max'; $A_i$max $= A_i$
2. Voting: For each landmark j in unlabeled data:
    • Pick most common identity for landmark j over the set of identities in $A_i$max $_{i=1...n}$
    • If 75% supermajority then the identity of landmark j is considered confident
3. Reiteration: Repeat steps 1–2 above with the modifications:
    • Substitute TPS warping using confident identities in place of pre-alignment using manual landmarks and GMCCPD
    • Confident identities in unlabeled data remain fixed and are not recomputed

## Algorithmic ablation results

To evaluate the contributions of distinct components of our method to performance we compare performance after ablating parts of the algorithm to our full algorithms' performance (71–78%). Ablating all of the unique components of our approach leaves the prior method of an LAP matching based on distance (after pre-alignment) to each labeled example followed by voting, this has accuracy of 43–66%. The most novel component of our system the co-optimization of unstable landmarks makes a significant contribution to accuracy; if we skip this step single cell accuracy ranges from 67% to 75%.

## Computational considerations

Our implementation has a significant runtime but could be significantly optimized if desired. Our co-optimization algorithm was implemented in Matlab. Alignment was performed on a Xeon Gold 6128 3.4 GHz CPU with 500 mb of memory (alignment is however not particularly memory intensive). Runtime was variable depending on the temporal stage and number of modifications to be considered but took about a day. The vast majority of run time is taken by the tens of thousands of iterations

of linear assignment implicit in the modification and constraint optimization process. Results are fairly insensitive to the number of constraint iterations used. Reducing iterations and total computation time by 3/4 results in only a small decrease in accuracy. Given the small number of datasets computation is not a major issue and reported timing and accuracy results reflect a large number of iterations run after initial landmarks and parameters were optimized using runs with a low number of iterations. A more efficient approximate LAP implementation would significantly decrease computational costs as over 95% of computational time is spent in this function. Runtime would also be significantly reduced by an explicitly parallelized implementation as alignment of individual labeled datasets, currently sequential, is completely independent; this was not attempted.

## Additional validation

To further assess our algorithm, we test its ability to name additional independent point clouds based on FM microscopy. In all experiments we use the same staging method, model, parameters, and same set of initial landmarks for pre-alignment as in EM experiments. Our additional test set contains 3 embryos imaged for 8 hr with a spinning disc confocal microscope. All embryos are lineaged and globally curated through 1.5-fold to ensure accurate cell identities for validation. Note that while our model is based on light sheet FM on uncompressed embryos, these additional test data sets are compressed while imaged, representing a challenge to alignment at least as significant (though distinct) as distortions from AT EM. We blind ourselves to identities except for the exact same pre-alignment landmarks used in the EM. Alignment is performed on a point cloud taken from the equivalent stage of each data set and accuracy is then assessed over all cells, since a complete ground truth is available. Result accuracy is comparable to EM and presented in *Supplementary file 1f*.

## Manual curation

### Manual curation of cell identities

A significant number of cells in each EM dataset were manually curated to establish their identities at the tissue or individual cell level. Tissue level annotation was created completely manually, moving one by one through each cell and annotating cells whose type could be readily assigned visually. This was possible based on overall morphology for the majority of pharynx, epidermis, gut, and body wall muscle cells, as well as some neurons.

### Single-cell annotation

Verified individual cell identities fall into two conceptually distinct categories: a. landmarks used in pre-alignment b. additional curated identities used in evaluation and for navigation.

The majority of pre-alignment landmarks are individual cells within readily identifiable tissues such as the gut, body wall muscles, and epidermis. The individual identities of cells within these tissue blocks can be unambiguously established by counting cells along the body axes within the block of tissue types. We found this most easily done by visualizing the cell centroids in 3D within WormGUIDES for comparison to a labeled data set. First, we identify tissues in the EM images and mark these cell centroids in a 3D visualization. Then, taking the gut as an example we count our way down the AP axis in the 3D model, identifying which gut nucleus centroid is which gut cell by comparing position to an identical 3D centroid visualization of a FM data set with all identities. In addition, there are a few landmarks which have dramatically distinct appearances (such as the excretory cell) or occupy highly distinctive positions on the dorsal, ventral or lateral midlines, or both (such as AVL precisely on the ventral midline, directly anterior of the excretory cell, and ALA, on the midline, and sharing an elongated left-right shape with its sibling RMED). These landmarks are very difficult to confuse with any other cell and are used as initial landmarks.

Additional curated identities for evaluation were first established for highly distinctive cells not used in pre-alignment but identifiable with the same procedures described above. Then, moving essentially in a ring out from these highly reliable landmarks we examine labeled and unlabeled data sets as 3D point clouds and identify cells with consistent position relative to a landmark in labeled data sets and the corresponding cell in EM. At the stages examined we have found this kind of relative position to be very reliable, with all observed discrepancies being local displacements traceable to variability of nuclear position within extended cell bodies. These tentative positional identifications are further validated against EM morphology, tissue boundaries, and known neuronal appearance

at this developmental stage, when this is available from FM experiments. At the comma and 1.5-fold stages individual neuronal shape provides strong hints for identification. Identified patches are expanded checking for inconsistencies until all cells in a local patch are identified. Though user errors are possible initially, once a patch of identities is extended sufficiently to span the area between highly distinctive cells/tissues, and all cells within an area are accounted for there is relatively little opportunity for human error. Total time spent on annotation varied between a few days for the least annotated data set and a bit more than a week for the best annotated comma data set.

### Neuronal outgrowth

Assigning cell identities in FM: In all FM data the identities of cells were determined via lineage tracing (see details in Section 2 above). Individual embryos were lineaged to identify individual cells or small clusters of cells expressing the membrane bound GFP transgenes detailed in *Supplementary file 1d*. Note that due to mosaicism the outgrowth dynamics of single cells can sometimes be assessed when expression in a region in potentially broader. Expressing cells once identified have their initial outgrowth times recorded. Assessing initial outgrowth in FM: Neurite initial outgrowth was defined in time lapse FM recordings as the first frame in which a distinct outgrowth was visible on a neuron cell body. This was further formalized as a protrusion reaching, and continuing to reach in future, in the direction of final outgrowth. This is distinct from possible earlier contractility of the cell surface which does not display this directedness. In small clusters of expressing cells, the cell used to name the cluster was confirmed to have an elongated shape aligned with nearby expressing cells, suggesting the outgrowth observed comes from multiple cells sharing the same orientation. Initial outgrowth timings from distinct embryos were aligned to a common temporal reference frame based on a linear normalization using cell division timings as landmarks (*Barnes, 2020*).

Assigning cell identities to neurites in EM: To identify individual neurite cross-sections labeled in figures, we manually traced neurites back to a cell body. Once neurites were traced back to cell bodies, nuclei were identified with the procedure described in section Additional validation above.

Assessing initial outgrowths in EM: Every cell in the vicinity of the ring was assessed for visible neurite outgrowth (though not all cells are pictured in *Figure 4e*). A cell was scored as broken out if it contained a significantly elongated structure extending in the direction expected from adult morphology. There is some ambiguity when observing static images between initial outgrowths and cells with elongated tear drop shapes that might precede outgrowth. The examples discussed have neurite outgrowths long enough to be unambiguously neurites, rather than tapered cell bodies. Other more ambiguous cells, other sublaterals, are arguable and may or may not have neurites at the bean stage. Temporal staging of samples was performed as described in section Temporal staging above.

## Acknowledgements

Thanks to Bruno Humbel, Jean Daraspe, Helmut Gnaegi, Tilman Franke for advice and assistance with EM, Chris Brittin, Li Fan, Kris Barnes and all the members of the WormGUIDES consortium for advice regarding *C. elegans* neurodevelopment. Particular thanks to Chris Brittin, Daniel Colón-Ramos, Hari Shroff, Mei Zhen, and Ryan Christensen for advice and comments on the manuscript.

## Additional information

### Funding

| Funder | Grant reference number | Author |
| --- | --- | --- |
| National Institutes of Health | R01GM097576 | Zhirong Bao |
| National Institutes of Health | R24OD016474 | Zhirong Bao |
| National Institutes of Health | P30CA008748 | Zhirong Bao |
| Chan Zuckerberg Initiative | 2019-198110 (5022) | Anthony Santella |

| Funder | Grant reference number | Author |
|---|---|---|
| Faculty of Biology and Medicine of the University of Lausanne | | Irina Kolotuev Caroline Kizilyaprak |

The funders had no role in study design, data collection and interpretation, or the decision to submit the work for publication.

## Author contributions

Anthony Santella, Conceptualization, Investigation, Methodology, Software, Visualization, Writing - original draft, Writing - review and editing; Irina Kolotuev, Conceptualization, Data curation, Investigation, Methodology, Resources, Supervision, Writing - original draft, Writing - review and editing; Caroline Kizilyaprak, Data curation, Methodology, Resources, Writing - review and editing; Zhirong Bao, Conceptualization, Funding acquisition, Investigation, Methodology, Resources, Supervision, Writing - review and editing

## Author ORCIDs

Irina Kolotuev [ID] http://orcid.org/0000-0003-1433-8048
Zhirong Bao [ID] http://orcid.org/0000-0002-2201-2745

## Decision letter and Author response

Decision letter https://doi.org/10.7554/eLife.77918.sa1
Author response https://doi.org/10.7554/eLife.77918.sa2

# Additional files

## Supplementary files

• Supplementary file 1. Additional summary tables detailing EM imaging conditions, EM and FM landmark properties, alignment accuracy, strains used in FM imaging, annotation extent and landmarks used to initialize alignment.

• Transparent reporting form

## Data availability

EM data has been made available on WebKnossos, and source code made available on https://github.com/zhirongbaolab/CellIDAlignment, (copy archived at swh:1:rev:7d5f9853cfc6f8def19e02a0a22a22a2c519e2ac).

The following datasets were generated:

| Author(s) | Year | Dataset title | Dataset URL | Database and Identifier |
|---|---|---|---|---|
| Santella A, Kolotuev I, Kizilyaprak C, Bao Z | 2021 | C. elegans bean stage EM | https://wklink.org/1322 | webKnossos, 1712,526,428,0,2.2,576 |
| Santella A, Kolotuev I, Kizilyaprak C, Bao Z | 2021 | C. elegans comma stage EM | https://wklink.org/1623 | webKnossos, 399,796,779,0,4.716,424 |
| Santella A, Kolotuev I, Kizilyaprak C, Bao Z | 2021 | C. elegans 1.5-fold stage EM | https://wklink.org/3489 | webKnossos, 940,634,77,0,1.129,249 |
| Santella A, Kolotuev I, Kizilyaprak C, Bao Z | 2021 | C. elegans two-fold stage EM | https://wklink.org/7824 | webKnossos, 1173,591,151,0,0.621,1130 |

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
