## [Editor Report]

This paper very nicely tackles a methodological problem in aligning different types of datasets (EM and light microscopy) to image embryonic nervous system development in the nematode *C. elegans*. The paper is important from a methodological standpoint, and also provides novel insights into nervous system development that will be of general interest.

---

## [Decision Letter]

**Decision letter after peer review:**

Thank you for submitting your article "Cross-modality Synthesis of EM Time Series and Live Fluorescence Imaging" for consideration by *eLife*. Your article has been reviewed by 3 peer reviewers, and the evaluation has been overseen by a Reviewing Editor (Oliver Hobert) and Claude Desplan as the Senior Editor. The following individual involved in the review of your submission has agreed to reveal their identity: Andrew D Chisholm (Reviewer #1).

All three reviewers agree that this is very nice work that we would like to see published in *eLife*. However, all three reviewers feel that the manuscript would very much profit from significant editorial revisions that emphasize the biological findings more and also contextualizes a number of these findings more effectively. You are also much encouraged to move material from the Supplement into the Main text, particularly those figures that provide novel biological insight. You will find below a quite extensive list of suggestions for editorial changes and we encourage you to consider them well.

*Reviewer #1 (Recommendations for the authors):*

Some issues in the presentation of the manuscript could be improved prior to publication. There appears to be mis-numbering of the supplemental figures and/or their callouts in the text. Although I have a theory as to the correct numbering, this created considerable confusion in the review.

The cell identification in the EM datasets is a combination of manual identification and automated identification based on the alignment algorithm. The manual identification process could be clarified, as it is extensive (140-240 manual IDs per data set). Table S5 summarizes some 'landmarks' used in pre-alignment, could the authors expand on what makes these neurons or other cells 'distinctive/distinctively shaped'? This table might be of more utility if fully written out at the single-cell level. It would be worth explaining what constitutes the 'ground truth' for these manual IDs.

Presentation and wording:

(1) The title states 'cross modality synthesis' of EM and FM imaging, however elsewhere it may be more accurately described as 'image alignment' or 'cross modality analysis'.

(2) Why is the EM data a 'pseudo time series? Pseudotime implies that the time dimension is inferred from non-temporal data, yet the EM data form a sparse time series (4 time points, though described as 'dense' in results).

(3) What is the definition of 'breakout' in the context of neurite growth?

(4) The *C. elegans*-specific term 'hypoderm' is not widely used; for a general audience 'epidermis' can be used as it means the same thing.

*Reviewer #2 (Recommendations for the authors):*

Strengths

– This is a first attempt for a combined study of *C. elegans* embryos by fluorescent and electron microscopy, by aligning – registering images without the use of correlative electron microscopy.

– This study introduces a landmark-based alignment assigning identities from a set of labeled data to unlabeled data, with co-optimization of unstable landmarks contributing to accuracy.

– This work provides a comprehensive analysis that largely supports and sometimes expands on previously studied processes of neuropil and sensory channel formation in *C. elegans* embryos.

– The authors provide publicly available data of their EM sections in webKnossos open resource, which is provided in amenable form can be very advantageous for the community.

Weaknesses

– The technical strength of the approach compared to others is not sufficiently demonstrated. There is a sparse actual assessment of the method's robustness/variability across samples and in comparison to other methods, in terms of data acquisition/analysis. Including comparative analysis of EM sample replicates for each time point, and comparisons across alternative methodologies (algorithms, electron microscopy) will allow substantiating the approach's superiority and its feasibility for future large-scale use, suggested by the authors in the study.

– Regarding data accessibility, the study sometimes lacks a detailed, comprehensive presentation of data acquisition, analysis, and underlying assumptions/caveats. This refers to instances such as the use of landmark fluorescent markers with multicellular expression to follow single-cell processes, the image registration/cell-identification using landmarks with expression patterns that are not presented in detail, explanations about process tracing and verification in EM sample analysis. The authors invite for community curation/ feedback of a publicly available EM resource, yet the provided resolution and formats (in terms of cell identification, tracing) may not facilitate curation.

– The study sometimes lacks comprehensive, inclusive contextualizing of new findings in relation to knowledge previously published by the authors and by other specialists. That refers to discussed developmental events of lineage progression guiding sample analysis, and findings of excretory pore and neuropil formation, pioneer-neuron concepts, amphid channel assembly.

I list my comments in order of appearance in the manuscript, which should be addressed for publication.

Suggestions for improved or additional experiments, data or analyses:

86-87 "As part of the effort to test and demonstrate the feasibility of EM time series acquisition[…]": The authors should discuss any potential variability/ differences arising from using different EM techniques to analyze different timepoints 1,2 and 3,4. Since the two techniques are used "to demonstrate feasibility" it appears key to discuss (later in the manuscript) the comparison of the two approaches in terms of their use in such project, i.e. effort/ implementation of sample preparation/data analysis, the influence of resulting resolution/stitching to cell identity registration, other pros/ cons. More importantly, if a single embryo is analyzed per timepoint, this would be one of my main critiques of the manuscript. At least 2 samples per timepoint will allow assessing the method's feasibility, robustness, reproducibility, variability.

92 "sparsely expressed cell membrane marker […] (Figure 1b, S1)": For the detailed understanding of *C. elegans* specialists, the authors should provide in supplementary data the exact lineage-derived expression pattern of used markers similarly to the Lineagomics project (https://epic.gs.washington.edu/), or else cite studies that provide these.

160-162 "Correlated cell identities […] using individual cells/nuclei as landmarks": It appears key that the authors present the variability in cell/nuclei positioning across embryos (and for different cell types) used for FM-lineaging. Especially given the use of these positions as landmarks on a single EM sample per timepoint.

165 "Based on timing of known developmental events in the FM series, the aligned EM data are timed": What are the known developmental events? This is not adequately or comprehensively explained. The authors should expand as necessary for a broader audience.

168 "We assess the accuracy of the correlated cell identities": Here and elsewhere in the manuscript, it appears important to compare the accuracy of cell identification and process tracing to other techniques of identification in electron microscopy studies such as CLEM, connectomics studies, or studies of EM-FM registration in other organisms (i.e. in Vergara, 2021 where nuclear-alignment results in 99.0% accuracy and stepwise image registration uses STAR Methods).

187-190 Figure 3g: "auto-fusion of the excretory canal, duct, pore and gland cell.": The images and notions concerning this topic and data should be explained in detail. Text and images lack context and details necessary even to *C. elegans* readers (unless they are specialists of this structure). Relevant studies should be discussed and cited (Soulavie et al., 2017,PMID: 29717108).

201 "These results are deposited on webKnossos.": The accessibility of the data is commendable. The authors are encouraged to explain what possibilities of annotations/ data-browsing are allowed in the interface. Given the study emphasizes describing fine cell morphologies, providing images with higher resolution and traces of the studied cells/processes across sections would allow better data visualization and integrating community work/ feedback.

215 "appear around the bean stage and a ring around the pharynx becomes visible by the comma stage": Earlier paper showing this should be cited here (Rapti et al., 2017, PMID: 28846083).

246 Figure 4c: "NR showing pioneers, entry points, the approximate extent at bean and comma."

Given there is one schematic in 4c, it is unclear if it corresponds to bean or comma. This should be explained. Entry points/growth can be different between bean and comma stages, being 25min apart.

246,262, 272 The authors often use the term pioneers to denote early growing neurites in the specific entry points. However, in other neuroscience studies including *C. elegans* studies (also cited by the authors), the term pioneers is used for early entering axons that are functionally important for entry of others. AVJ-AVD (referred to as supralateral commissure pioneers) ASH, RIB, and AWC (referred to as amphid commissure pioneers) have not been shown functionally important. The authors should refer to early entering neurites and not pioneers. Otherwise, pioneer roles should be established with ablation/functional studies, or other studies showing functional importance should be cited accordingly.

252 "Figure 4e: Each dot represents a single neuron or small cluster of neurons. (lim-4:red, unc-86:green, zag-1:blue, ttx-3:magenta, egl-13:cyan, cnd-1:black, ceh-37:gray, ceh-10:purple).": The expression patterns of the used labels are missing here, or please indicate the published source.. Given their expertise in lineaging, it is important that the authors provide (supplementary) the lineage-related expression pattern of used markers as in Lineagomics project (https://epic.gs.washington.edu/). Secondly, at least some of these reporters do not label single cells (i.e. mgIs18, Table S4 labels AIY but also SMDD as per Bertrand and Hobert, 2009; Rapti et al., 2017). Moreover, many of these labels used (Table S4) appear to be non-integrated arrays (egl-13, ceh-37, zag-1, lim-4?) and thus are prone to mosaicism. It is unclear if/how the breakout times of specific neurons are identified in the single-cell resolution. In FM, are these identified by lineaging these markers and identifying processes in single-cell resolution? In EM, are these identified through registration of cell identities and EM-tracing of single-cell axons? The authors should explain. Replying to the above is key to allow contextualizing the findings with other published information on neuropil assembly, for a better understanding of the reader.

255 and 260 "Figure 4f […] Number of neurites visible in g across the EM series.": The authors should explain in detail what each dot represents in EM timepoints: is this the total number of neurites organized in bundles or also includes single neurites visible across EM sections? Does it refer to neurites across the whole embryo or the ones contributing to the NR neuropil? In the latter case, are the NR-contributing neurites defined by cell identities or by tracing in the NR?

279-280 "a sensory structure that is open to the environment […] The dendrites grow via a distinctive mechanism, a collective retrograde extension": Authors should cite earlier papers demonstrating this structure by EM, including in embryos! (Ward et al., 1975,PMID: 1112927; Oikonomou, et al., PLoS Biol. 2011) (Low et al., Development, 2019, PMID: 30683663). The first paper demonstrating the mechanism of retrograde extension should also be cited (Heiman, Shaham, 2009, PMID: 19344940). I find it puzzling that these are missing.

281 “Formation of the embedded opening and elaboration of dendrite morphology occur in embryogenesis but are less understood.”: In fact, mechanisms of the channel’s formation are significantly studied. A series of relevant publications investigate this process, with mechanistic findings (Perens and Shaham, 2006 PMID: 15935778; Oikonomou, et al., PLoS Biol. 2011 PMID: 21857800; Oikonomou et al., 2012 PMID: 22138055; Wang et al., 2017 PMID: 28803967; Bacaj, Lu, Shaham, 2008 PMID: 18245347; Low et al., 2019, PMID: 30683663). These should be discussed and cited.

361 "Last but not least, our post hoc correlation of EM and FM data provides a useful alternative to true correlative EM. Correlative EM is a powerful, but complex to implement […]": For this first demonstration of cell identification with post hoc FM-to-EM correlation, it appears important to validate the method with CLEM of one sample. An embryo expressing a cell landmark marker with sparse labeling (i.e. Pttx-3) could be prepared by correlative EM so as to identify the labeled cells (AIY,SMDD) by CLEM labeling and by the new co-optimization algorithm. This will allow evaluating the accuracy of nuclei estimation. It should be hopefully feasibly, given the author's previous expertise/use of Correlative EM (Kolotuev et al., 2009, PMID: 19807690; Kolotuev et al., 2012,PMID: 22857930; Burel et al.,2018, PMID: 29802150).

339 "While our study is focused on the wild type, one could use live FM to build the appropriate ensemble models to analyze mutants": Using this methodology to study mutants is an exciting perspective! Yet, WT-mutant comparisons require analysis of more than a single individual per genetic condition. The study in its current form does not allow to evaluate the feasibility of larger-scale usage. Also, EM analysis of one sample per condition/timepoint does not inform on possible variability across samples of one timepoint. Performing EM analysis of a second sample for at least one timepoint per technique will allow addressing the issues.

341-343 "We present the image data and identity annotation of our correlated EM series as an accessible public resource […] seed of a community effort that we hope will fully validate annotation"

The public accessibility of the resource is commendable. It would be important to understand what type of data-browsing and annotations are allowed by the interface and the resolution of the data provided. In order for effective browsing, 3D data could be available with the identified cell identities annotated on the data. Process tracing of the identified cells could also be provided on the browsed/ downloadable data. In order for validation and future use of this resource by the community, the data would need to be provided in a downloadable format with the required resolution.

354 "The data-driven modification in the co-optimization algorithm further facilitates such a vision for automated alignment and annotation of complex brain images.": It is unclear to the broader audience to which degree this is feasible, what advances the current method requires. The authors should explain what limitations or difficulties may be faced for the implementation of the approach in more complex images.

Figure S7d "socket cell over three successive frames in the same cnd-1 promoter"

By Lineagomics (https://epic.gs.washington.edu/) cnd-1 is expressed in many more cells than the amphid socket, including cells partaking in the amphid channel (AMsh sheath, AWC, ASG). AMso and AMsh could be imaged distinguishably by performing mosaic experiments or photoconversion or using less broad promoters. If not, it is unclear how the authors distinguish in FM between processes of all these cells in Figure 5d and Figure 7d, given the FM resolution. Because of this, it is unclear what the FM imaging is adding to the EM data and conclusions. The authors should explain.

Recommendations and corrections for text and figures:

62 "EM studies [contributed to] principles of structural organization"

Despite this being a more comprehensive study of embryonic tissues by EM, few past studies in *C. elegans* embryos contributed principles of structural organization (Soulavie et al., 2017, PMID: 29717108, Rapti et al., 2017, PMID: 28846083, Low et al., 2019, PMID: 30683663). It appears appropriate that such studies are cited here.

98 “we use the nuclear positions, which are also annotated for the EM series, as the common reference (i.e., a Rosetta stone)”: Authors should expand and explain adequately/ comprehensively as necessary for a broader audience.

185 Aligning color-codes to use the same in Figure 3e and b-c-d, will allow the following cell identities.

189 EM images can be shown both with and without color-coding to better visualize cell fusion.

213 “The NR emerges in the later half of embryogenesis”: Earlier paper showing this should be cited here (Rapti et al., 2017, PMID: 28846083).

217 “even neurite topography through combinations of two-color imaging”: Earlier studies with *C. elegans* two-color FM imaging of neurites (including in the embryo!) can be cited (Heiman, Shaham, 2009, PMID: 19344940, Rapti et al., 2017, PMID: 28846083, Moyle et al., 2020, PMID: 33627875)

220 “We scan through EM data to identify neurites, and backtrace them to soma”: The authors should explain the exact approach used for process tracing (manual, semi-/ automatic, etc).

240-250 Figure 4b-4e. Color-coding is not consistent between models, FM, EM images, i.e. sublateral pioneers are red in Figure 4b EM but light blue in Figure 4c, and Plim-4 which labels sublateral pioneers is red in Figure 4e. Consistent color-coding would help the reader.

Figure 4c, S5a It is unclear if the drawn arrows’ length is proportional to the actual neurite total growth (across EM sections of the same embryo)? For example, what is the relative length of SIA/SMD and ASH, in the model (Figure S5a) and in the actual EM (Figure S5f,g)?

268-271 “comparing the left- and right-side behavior of the first cells to grow into the NR, namely SIADL/R and SMDDL/R, suggests strong symmetry in behavior. Both sides have broken out at the bean stage, with similar-sized outgrowths […] their tips meet precisely beneath the midpoint of the ALA […].”

This is in line with previous studies, indicating such symmetries and ALA’s position, these studies should be cited (Rapti et al., 2017, PMID: 28846083; Insley and Shaham, 2018, PMID: 29590193; Moyle et al., 2021, PMID: 33627875). In order to establish symmetry beyond previous observations, it appears key to providing quantitative data, calculating the length of bilateral processes by tracing them in EM series.

282 The first paper demonstrating this dynamic process should be cited (Heiman, Shaham, 2009, PMID: 19344940).

Figure S5B Shading of EM figure in Box b doesn’t allow neurite visualization. Also (box B), how it is distinguished whether neurons extend neurites or have elongated cell bodies? It would help to provide the EM images with and without color-coding (and larger sizes) to facilitate visualization.

274 “It is not clear […] if there is signaling to coordinate between the two sides.”

Some mechanisms related to the regulation of left-right symmetry of axon guidance are studied in Grossman, Giurumescu, Chisholm, 2013, PMID: 23979582. The paper should be cited.

Figure S3, S4, S5, S6, S7 are mislabeled, 2 figures are abelled S3 and legends do not correspond. Also, scale bars are missing in many supplementary figures and when present, there is no reference of scale bar size in the legend. These should be corrected accordingly.

411 Some alignment of 1.5/2fold AT-EM data (available in webKnossos) is “jittery”, may any additional image registration help eliminate this, for the better visualization of data?

*Reviewer #3 (Recommendations for the authors):*

The method is well-executed and thoroughly explained, and this manuscript shines in the temporal and spatial precision by which developmental processes are uncovered and described as a result. However much of these biological discoveries were buried in the supplementals. To create more relevance and excitement for this journal’s audience, the authors should re-write parts of the manuscript, with a stronger emphasis on what the limitations of current imaging methods are, what cross-modality analyses improve upon, and particularly by bringing focus to the biological discoveries and putting them into the main article.

In addition to the inherent challenges of EM-FM cross-comparison, in this article, the cross-modality alignment method overcomes two distinct issues: the challenges associated with manual annotation of embryo developmental EM series; as well as integration of spatial resolution of EM with the temporal resolution with FM. These concepts are briefly mentioned in the introduction and throughout the article, but do not become apparent to a non-specialist reader until the end. These current challenges should be further emphasized in the introduction when describing these imaging methods to highlight the benefits of the analysis method presented.

Note: supplemental figures are mis-numbered (there are two Figure S3s, and the following figures do not match figure captions). I will refer to supplemental figures as they are numbered in the figure captions.

Most of Figures S5, S6 and S7 should be integrated into or added as additional main figures. These data beautifully highlight both the temporal and spatial resolution achieved by the cross-modality method used.

How is “adjacency” defined between nuclei that are not physically touching? Are there specific parameters for what distance between cell nuclei counts as adjacent?

Line 236: what is meant by “breakouts observed in the EM at the comma stage span an hour?”. Is this referring to lack of temporal resolution? This statement is confusing because Figure 4e and Table S4 referenced doesn’t seem to show this to be the case.

---

## [Author Response]

Reviewer #1 (Recommendations for the authors):Some issues in the presentation of the manuscript could be improved prior to publication. There appears to be mis-numbering of the supplemental figures and/or their callouts in the text. Although I have a theory as to the correct numbering, this created considerable confusion in the review.

Apologies for the supplemental figure mis-numbering which has been corrected. We have also revised and attaching supplemental figures to the primary figures to better integrate them into the main text.

The cell identification in the EM datasets is a combination of manual identification and automated identification based on the alignment algorithm. The manual identification process could be clarified, as it is extensive (140-240 manual IDs per data set). Table S5 summarizes some ‘landmarks’ used in pre-alignment, could the authors expand on what makes these neurons or other cells ‘distinctive/distinctively shaped’? This table might be of more utility if fully written out at the single-cell level. It would be worth explaining what constitutes the ‘ground truth’ for these manual IDs.

We have revised Methods with a new section 4 focused on Manual Curation to clarify identity validation. We clarify that manual annotation of EM includes two conceptually distinct kinds of annotation. A. Landmarks used in prealignment, uniquely identifiable relatively trivially by appearance (and in some cases by counting along the AP axis within an easy to identify class) or by adjacency to a trivially recognizable cell. B. Curated cell identities recognized either in the same manner, or with greater effort by counting outward in rings from initial landmarks and confirming by morphology or exhaustive accounting. We have significantly expanded on our description in methods of how these identities are established. In addition, we also revise our presentation of these two forms of annotation. Landmark Table S5 (now referenced on reformatting as Supplementary File 1g) has been supplemented with a new Supplementary File 1e providing a more detailed summary of what cells are annotated. A list of confirmed and predicted identities for all cells remains in supplemental data and available on webKnossos.

Presentation and wording: (1) The title states ‘cross modality synthesis’ of EM and FM imaging, however elsewhere it may be more accurately described as ‘image alignment’ or ‘cross modality analysis’.

We retain ‘cross modality synthesis’ in the title and have adjusted text to be more consistent in using this term to describe the synthesis of the EM and FM data sets in the biological demonstrations, as distinct from the ‘alignment’ process itself which enables that synthesis, for which we now consistently use the term alignment.

(2) Why is the EM data a ‘pseudo time series? Pseudotime implies that the time dimension is inferred from non-temporal data, yet the EM data form a sparse time series (4 time points, though described as ‘dense’ in results).

We had used pseudo to differentiate our EM series, assembled from multiple individuals, from a true time lapse data set, but have removed the modifier as well as dense as they seem more likely to confuse rather than clarify our meaning.

(3) What is the definition of ‘breakout’ in the context of neurite growth?

By breakout we mean the event of initial outgrowth of a neurite from a neuronal cell body. We have revised the wording to “initial outgrowth”.

(4) The C. elegans-specific term ‘hypoderm’ is not widely used; for a general audience ‘epidermis’ can be used as it means the same thing.

We have replaced hypoderm with epidermis throughout for clarity.

Reviewer #2 (Recommendations for the authors):Strengths– This is a first attempt for a combined study of C. elegans embryos by fluorescent and electron microscopy, by aligning – registering images without the use of correlative electron microscopy.– This study introduces a landmark-based alignment assigning identities from a set of labelled data to unlabeled data, with co-optimization of unstable landmarks contributing to accuracy.– This work provides a comprehensive analysis that largely supports and sometimes expands on previously studied processes of neuropil and sensory channel formation in C. elegans embryos.– The authors provide publicly available data of their EM sections in webKnossos open resource, which is provided in amenable form can be very advantageous for the community.Weaknesses– The technical strength of the approach compared to others is not sufficiently demonstrated. There is a sparse actual assessment of the method’s robustness/variability across samples and in comparison to other methods, in terms of data acquisition/analysis. Including comparative analysis of EM sample replicates for each time point, and comparisons across alternative methodologies (algorithms, electron microscopy) will allow substantiating the approach’s superiority and its feasibility for future large-scale use, suggested by the authors in the study.– Regarding data accessibility, the study sometimes lacks a detailed, comprehensive presentation of data acquisition, analysis, and underlying assumptions/caveats. This refers to instances such as the use of landmark fluorescent markers with multicellular expression to follow single-cell processes, the image registration/cell-identification using landmarks with expression patterns that are not presented in detail, explanations about process tracing and verification in EM sample analysis. The authors invite for community curation/ feedback of a publicly available EM resource, yet the provided resolution and formats (in terms of cell identification, tracing) may not facilitate curation.– The study sometimes lacks comprehensive, inclusive contextualizing of new findings in relation to knowledge previously published by the authors and by other specialists. That refers to discussed developmental events of lineage progression guiding sample analysis, and findings of excretory pore and neuropil formation, pioneer-neuron concepts, amphid channel assembly.

We thank the reviewer for their comments, in response to which we have further contextualized our findings, clarifying the methods used in our biological case studies and the relationship of these studies to prior literature. We have also elaborated on the performance of our computational method and implications for the study of spatio-temporal consistency in development. Details follow.

I list my comments in order of appearance in the manuscript, which should be addressed for publication.Suggestions for improved or additional experiments, data or analyses:86-87 “As part of the effort to test and demonstrate the feasibility of EM time series acquisition[…]”: The authors should discuss any potential variability/ differences arising from using different EM techniques to analyze different timepoints 1,2 and 3,4. Since the two techniques are used “to demonstrate feasibility” it appears key to discuss (later in the manuscript) the comparison of the two approaches in terms of their use in such project, i.e. effort/ implementation of sample preparation/data analysis, the influence of resulting resolution/stitching to cell identity registration, other pros/ cons). More importantly, if a single embryo is analyzed per timepoint, this would be one of my main critiques of the manuscript. At least 2 samples per timepoint will allow assessing the method’s feasibility, robustness, reproducibility, variability.

Regarding lessons concerning EM techniques from our efforts, we note the EM methods used are well established and cited where appropriate, a full technical comparison is beyond the scope of this paper. However, we have added a section to Discussion further contextualizing the pros and cons of the two techniques in the context of the worm embryo and cross-modality analysis.

Regarding the need for multiple data points per stage there are two goals served by replication: I. Establishing the reproducibility of our alignment process itself. II. Investigating the consistency of the biological processes observed.

Regarding I, we have added additional quantification of our alignment method’s accuracy in predicting cell identity. These are derived by aligning additional independent FM embryos against the same three FM embryos used to name the EM data. Since data sets are represented in all cases by abstract point clouds this experiment provides a valid quantitation of our algorithm’s success in identifying cells in a larger set of embryos. Furthermore, the cell names in these independent FM data sets are derived from lineaging and as such provide an orthogonal ground truth, which is more certain and definitive than the manually curated cell identities in the main EM alignment experiments. Finally, it is worth nothing that these FM embryos were imaged compressed between coverslips (unlike the 3 FM embryos used as the model or the EM embryos). The compression creates nonlinear deformations throughout the embryo and success in identifying cells in these compressed embryos further demonstrates the robustness of our method. These additional experiments are now mentioned in results, described in Methods, and performance in them added as Supplementary File 1f.

Regarding II, the biological consistency of events, we believe that in terms of temporal consistency as well as coarse level spatial consistency (e.g. cell soma positions), one should turn to FM for data coverage that cannot be afforded by EM in the foreseeable future. In terms of fine spatial consistency, we agree that one wants additional EM experiments, and we are in the process of doing this, but we also feel that this is outside the scope of the current study, which seeks to establish the feasibility of the approach and provide an initial time series for the community.

92 “sparsely expressed cell membrane marker […] (Figure 1b, S1)”: For the detailed understanding of C. elegans specialists, the authors should provide in supplementary data the exact lineage-derived expression pattern of used markers similarly to the Lineagomics project (https://epic.gs.washington.edu/), or else cite studies that provide these.

The global expression pattern of these markers is not used to ID cells, and in some cases has not been fully established. Rather, in every case, FM identities are established by lineaging expressing cells in the specific data set utilized. We have clarified this in the new Methods section 4.2.

160-162 “Correlated cell identities […] using individual cells/nuclei as landmarks”: It appears key that the authors present the variability in cell/nuclei positioning across embryos (and for different cell types) used for FM-lineaging. Especially given the use of these positions as landmarks on a single EM sample per timepoint.

We agree that characterization of variability is important, and a topic of great interest to us. Variability is a complex topic with spatial and temporal aspects and many perspectives on whether, for example, cell position, at each developmental stage is unimodal and can be meaningfully represented by a mean. The field of development has struggled with this problem and to our knowledge, not come up with succinct or definitive answers. If this is possible, it will require more example data than we have currently available.

This said, in revisions we elaborate on ways that our computational approach might further an understanding of variability. We argue that variability in adjacency is a natural framework in which to characterize positional variability. While we acknowledge our adjacency measure is a computational approximation of the underlying reality of cell contacts, it has the same strong advantages here as in our alignment method such as simplicity and invariance to absolute position. Any other characterization of variability will have its own limitations and caveats.

While our alignment method uses only universally present adjacencies in alignment, there is significant, more granular variation in how consistently adjacencies are observed. We briefly present this adding an additional supplemental figure and supplemental data characterizing the quantitative variability in adjacency at each stage. We illustrate that this model of position can be used to characterize changes between stages and to capture groups of cells with related behaviors over time. This approach may serve as a promising framework for future work.

165 “Based on timing of known developmental events in the FM series, the aligned EM data are timed”: What are the known developmental events? This is not adequately or comprehensively explained. The authors should expand as necessary for a broader audience.

We further clarify our approach for temporal staging in methods section 3.2 and now explicitly reference this in the main text. Briefly, we use overall elongation at comma and 1.5 fold stages. We supplement this with a key cell division at the earlier bean stage where body morphology is less reliable. We also note that though choosing the correct temporal window for alignment is critical, it is not necessary to establish timing to within a few minutes, but rather only to pick a reasonable model (assembled over a 13 min sliding window) to align against.

168 “We assess the accuracy of the correlated cell identities”: Here and elsewhere in the manuscript, it appears important to compare the accuracy of cell identification and process tracing to other techniques of identification in electron microscopy studies such as CLEM, connectomics studies, or studies of EM-FM registration in other organisms (i.e. in Vergara, 2021 where nuclear-alignment results in 99.0% accuracy and stepwise image registration uses STAR Methods).

We agree accuracy is critical and have added additional evaluation of the accuracy of our alignment method in establishing *C. elegans* cell identities, as detailed above.

Regarding other alignment methods the initial mansucript discusses the performance of alternative computational algorithms. We present their reported accuracy for comparison given it would be difficult, and not necessarily fair to adapt each method to our data given different models and assumptions.

Overall, we would argue that manual identification based on morphology and position is the near universal default approach and there are few other baselines to compare to. This includes Vergara 2021 where cell types are identified by morphology even in the context of extensive aligned expression data. The 99% accuracy in this paper cited is for nuclear detection rather than identity alignment. We did not find any results on identity alignment in that paper.

CLEM could in theory provide ground truth for a subset of cell identities, albeit at the cost of a technically complex method that might compromise the quality of ultrastructure imaging. However, we feel our additional computational experiments provide a better assessment of alignment quality. This is significantly more scalable and allows validation of every cell’s prediction as opposed to a small subset of cells.

187-190 Figure 3g: “auto-fusion of the excretory canal, duct, pore and gland cell.”: The images and notions concerning this topic and data should be explained in detail. Text and images lack context and details necessary even to C. elegans readers (unless they are specialists of this structure). Relevant studies should be discussed and cited (Soulavie et al., 2017,PMID: 29717108).

We have added the reference requested and edited the text here to briefly describe the events and to clarify the context that our point is not to investigate this event, but rather to demonstrate the ability to pick a subtle developmental event, and follow it clearly over time in our series. The example provides a demonstration of how the data can be used, including in the field of cell biology. As we don’t have significant new findings to report here discussion remains compact.

201 “These results are deposited on webKnossos.”: The accessibility of the data is commendable. The authors are encouraged to explain what possibilities of annotations/ data-browsing are allowed in the interface. Given the study emphasizes describing fine cell morphologies, providing images with higher resolution and traces of the studied cells/processes across sections would allow better data visualization and integrating community work/ feedback.

We expand briefly on our description of the webKnossos interface, highlighting relevant functionality, the ability to browse images and annotation together and for users to ‘fork’ and re-annotate the data set. We also expanding on our hope to extend annotation with contributions from the community.

Every method of data distribution has its limitations, but we hope webKnossos will maximize the ability of users to quickly find areas of interest, create, and share annotations.

We have also added a note to the manuscript asking readers to contact the authors for direct access to raw image data.

215 “appear around the bean stage and a ring around the pharynx becomes visible by the comma stage”: Earlier paper showing this should be cited here (Rapti et al., 2017, PMID: 28846083).

We apologize and add this citation.

246 Figure 4c: “NR showing pioneers, entry points, the approximate extent at bean and comma.”Given there is one schematic in 4c, it is unclear if it corresponds to bean or comma. This should be explained. Entry points/growth can be different between bean and comma stages, being 25min apart.

We clarify the caption of Figure 4c. It now states that the first arrows in the amphid and sublaterals indicate first outgrowth at bean and the later paths and arrows (and the whole path on other neurons) represent approximate comma entry points and outgrowth extent. Entry to the ring proper has not occurred at bean, which represents the first neuronal outgrowths*.*

246,262, 272 The authors often use the term pioneers to denote early growing neurites in the specific entry points. However, in other neuroscience studies including C. elegans studies (also cited by the authors), the term pioneers is used for early entering axons that are functionally important for entry of others. AVJ-AVD (referred to as supralateral commissure pioneers) ASH, RIB, and AWC (referred to as amphid commissure pioneers) have not been shown functionally important. The authors should refer to early entering neurites and not pioneers. Otherwise, pioneer roles should be established with ablation/functional studies, or other studies showing functional importance should be cited accordingly.

We clarify this removing pioneer and replacing this with early, as our goal is to document first outgrowths along the path of the ring, rather than establish functional guidance ‘pioneering’.

252 “Figure 4e: Each dot represents a single neuron or small cluster of neurons. (lim-4:red, unc-86:green, zag-1:blue, ttx-3:magenta, egl-13:cyan, cnd-1:black, ceh-37:gray, ceh-10:purple).”: The expression patterns of the used labels are missing here, or please indicate the published source.. Given their expertise in lineaging, it is important that the authors provide (supplementary) the lineage-related expression pattern of used markers as in Lineagomics project (https://epic.gs.washington.edu/). Secondly, at least some of these reporters do not label single cells (i.e. mgIs18, Table S4 labels AIY but also SMDD as per Bertrand and Hobert, 2009; Rapti et al., 2017). Moreover, many of these labels used (Table S4) appear to be non-integrated arrays (egl-13, ceh-37, zag-1, lim-4?) and thus are prone to mosaicism. It is unclear if/how the breakout times of specific neurons are identified in the single-cell resolution. In FM, are these identified by lineaging these markers and identifying processes in single-cell resolution? In EM, are these identified through registration of cell identities and EM-tracing of single-cell axons? The authors should explain. Replying to the above is key to allow contextualizing the findings with other published information on neuropil assembly, for a better understanding of the reader.

In FM:

a. Identities are established by lineaging individual cells *in each dataset*, see response above and clarification in Methods.

b. Some points represent small clusters. We make this clearer in the manuscript as a caveat. We note, however, that in these (few) cluster cases two pieces of corroborative evidence exist. 1. We examine the cell soma shape – these have a tear drop shape indicating that the cell of interest appears to participate in the bundle of outgrowth near its pointed end. 2. Outgrowth for that specific cell was observed in EM around that time. This is clarified in Methods.

c. Mosaicism is present in these data sets. Because of this cell soma are confirmed as abelled in the specific data sets used and identity is established by lineaging, not expression status. In a number of cases loss of expression in neighboring, potentially expressing cells is used to achieve single cell resolution. This is clarified in Methods.

In EM:

Yes, identities are assigned by registration and then confirmed by morphology and relative position as described in expanded Methods.

255 and 260 “Figure 4f […] Number of neurites visible in g across the EM series.”: The authors should explain in detail what each dot represents in EM timepoints: is this the total number of neurites organized in bundles or also includes single neurites visible across EM sections? Does it refer to neurites across the whole embryo or the ones contributing to the NR neuropil? In the latter case, are the NR-contributing neurites defined by cell identities or by tracing in the NR?

We have clarified more explicitly that 4f is quantifying only neurites visible in the specific cross sectional location of the ring indicated.

279-280 “a sensory structure that is open to the environment […] The dendrites grow via a distinctive mechanism, a collective retrograde extension”: Authors should cite earlier papers demonstrating this structure by EM, including in embryos! (Ward et al., 1975,PMID: 1112927; Oikonomou, et al., PLoS Biol. 2011) (Low et al., Development, 2019, PMID: 30683663). The first paper demonstrating the mechanism of retrograde extension should also be cited (Heiman, Shaham, 2009, PMID: 19344940). I find it puzzling that these are missing.

We apologize for the omission of (Heiman, Shaham, 2009, PMID: 19344940 inadvertently deleted during initial manuscript revisions.

We have added Ward et al., here to provide a more comprehensive crediting of work defining the anatomy of the amphid structure. Low et al., is already cited in a what seems to us a more appropriate location in the successive sentence, in the context of the retrograde extension.

We have added Oikonomou et al., 2011 which we apologize for overlooking the more directly relevant context suggested below and the later relevant context of the essentially adult like structure being present in late embryogenesis.

281 “Formation of the embedded opening and elaboration of dendrite morphology occur in embryogenesis but are less understood.”: In fact, mechanisms of the channel’s formation are significantly studied. A series of relevant publications investigate this process, with mechanistic findings (Perens and Shaham, 2006 PMID: 15935778; Oikonomou, et al., PloS Biol. 2011 PMID: 21857800; Oikonomou et al., 2012 PMID: 22138055; Wang et al., 2017 PMID: 28803967; Bacaj, Lu, Shaham, 2008 PMID: 18245347; Low et al., 2019, PMID: 30683663). These should be discussed and cited.

In the text we clarify our statement to refer to limited knowledge of the early processes of formation for the structure of the sensory organ in the embryo, which we believe remains correct.

As noted above we have added Oikonomou 2011, and an additional publication we apologize for overlooking Nechipurenko 2017 to acknowledge prior knowledge regarding embryonic development of the overall sensory structure.

We have also added suggested citations to credit work characterizing necessary molecular machinery for amphid formation: Perens and Shaham 2006 Oikonomou 2012, Wang et al., 2017 for glial formation and Bacaj et al., 2008 for cilia formation.

Low 2019 is already cited in what we feel is a more appropriate context of the overall formation process.

361 “Last but not least, our post hoc correlation of EM and FM data provides a useful alternative to true correlative EM. Correlative EM is a powerful, but complex to implement […]”: For this first demonstration of cell identification with post hoc FM-to-EM correlation, it appears important to validate the method with CLEM of one sample. An embryo expressing a cell landmark marker with sparse abelling (i.e. Pttx-3) could be prepared by correlative EM so as to identify the abellin cells (AIY,SMDD) by CLEM abelling and by the new co-optimization algorithm. This will allow evaluating the accuracy of nuclei estimation. It should be hopefully feasibly, given the author’s previous expertise/use of Correlative EM (Kolotuev et al., 2009, PMID: 19807690; Kolotuev et al., 2012,PMID: 22857930; Burel et al.,2018, PMID: 29802150).

As we mention above we feel that we have addressed this issue with additional algorithm evaluation on FM data sets. While the experiment described would be possible, as we discuss above CLEM validation is neither necessary nor (in general) a scalable method of definitive cell identification. The effort of collecting a CLEM dataset seems out of scope in this project. This seems particularly true as cells can be directly identified manually with great certainty in an *in toto* 3D EM data set.

339 “While our study is focused on the wild type, one could use live FM to build the appropriate ensemble models to analyze mutants”: Using this methodology to study mutants is an exciting perspective! Yet, WT-mutant comparisons require analysis of more than a single individual per genetic condition. The study in its current form does not allow to evaluate the feasibility of larger-scale usage. Also, EM analysis of one sample per condition/timepoint does not inform on possible variability across samples of one timepoint. Performing EM analysis of a second sample for at least one timepoint per technique will allow addressing the issues.

Establishing our method for a mutant (similar to the WT) requires: lineageing a few embryos from FM, and generate an EM time series. This would be a qualitative step forward, because it is more challenging, in general, to interpret mutant EM cell identity.

We agree that this is perhaps only appropriate for high penetrant mutants, given the scale of labor involved in acquiring sufficient EM of low penetrance mutants. Scaling this up to many mutants is an ambitious and aspiring goal. We suspect this will require further advances in speed and automation of EM.

We elaborate on this briefly in the revised Discussion.

See point 1 above for discussion of WT replication.

341-343 “We present the image data and identity annotation of our correlated EM series as an accessible public resource […] seed of a community effort that we hope will fully validate annotation”The public”accessibility of the resource is commendable. It would be important to understand what type of data-browsing and annotations are allowed by the interface and the resolution of the data provided. In order for effective browsing, 3D data could be available with the identified cell identities annotated on the data. Process tracing of the identified cells could also be provided on the browsed/ downloadable data. In order for validation and future use of this resource by the community, the data would need to be provided in a downloadable format with the required resolution.

Please see point 7 above.

354 “The data-driven modification in the co-optimization algorithm further facilitates such a vision for automated alignment and annotation of complex brain images.”: It is unclear to the broader audience to which degree this is feasible, what advances the current method requires. The authors should explain what limitations or difficulties may be faced for the implementation of the approach in more complex images.

We have expanded our discussion to clarify this hypothetical application. We elaborate the aspect of this larger problem our approach solves (spatial alignment with variable landmarks) and what additional data (identity information, most likely molecular information from spatial sequencing or FM) and processing techniques would be necessary to achieve our vision of automated spatio temporal atlas construction.

Figure S7d “socket cell over three successive frames in the same cnd-1 promoter”By Lineagomics (https://epic.gs.washington.edu/) cnd-1 is expressed in many more cells than the amphid socket, including cells partaking in the amphid channel (Amsh sheath, AWC, ASG). Amso and Amsh could be imaged distinguishably by performing mosaic experiments or photoconversion or using less broad promoters. If not, it is unclear how the authors distinguish in FM between processes of all these cells in Figure 5d and Figure 7d, given the FM resolution. Because of this, it is unclear what the FM imaging is adding to the EM data and conclusions. The authors should explain.

Cnd-1 indeed labels a wide array of cells, including a number of amphids. These cells do not effect interpretation of the images:

Most of these cells are not primarily part of this context, overlap between amphid dendrites and the socket and sheath cells is minimal, while FM lacks the resolution to resolve the details of amphid dendrite and glial interaction, the glial cell bodies can be visualized without interference from the dendrite tips.

More concerning is the adjacency of the sheath and socket, both cnd-1 labeled. Changes in the socket, as highlighted in Fig5d and Figure 5 S1d can be visualized because of its approximately 1.5x greater brightness over the adjacent sheath cell. This makes it highly unlikely the signal visualized originates in the dimmer sheath cell. This is more apparent in the included supplemental video. In addition, we point out that this is corroboration of morphology seen in the EM (though easy to overlook without the dynamic context of FM, an example of what FM adds) and not stand-alone evidence.

We have attempted to clarify this in the figure caption.

Recommendations and corrections for text and figures:62 “EM studies [contributed to] principles of structural organization”Despite this being a more comprehensive study of embryonic tissues by EM, few past studies in C. elegans embryos contributed principles of structural organization (Soulavie et al., 2017, PMID: 29717108, Rapti et al., 2017, PMID: 28846083, Low et al., 2019, PMID: 30683663). It appears appropriate that such studies are cited here.

We agree, it was not our intention to imply no EM of the embryo has ever been useful previously and we have added a line here citing these as examples where insight has been gained from readily identified structures. We have also added Nechipurenko et al., 2017: https://pubmed.ncbi.nlm.nih.gov/28411364/, additional relevant embryonic EM work in our discussion of the amphid cilia.

98 "we use the nuclear positions, which are also annotated for the EM series, as the common reference (i.e., a Rosetta stone)": Authors should expand and explain adequately/ comprehensively as necessary for a broader audience.

We have rephrased and expanded here to clarify our approach which is further elaborated throughout the paper.

185 Aligning color-codes to use the same in Figure 3e and b-c-d, will allow the following cell identities.

We agree consistent colors are in general helpful, and we have attempted to achieve this.

In 3 B,D which use one consistent color scheme the goal is to highlight the distinct global positions of different tissue types. In 3e,f,g a different, but internally consistent color scheme is used to highlight contrast between adjacent cells. Given limited choices some colors are used in both of these necessary schemes, we have attempted to clarify the color logic in the revised figure caption.

189 EM images can be shown both with and without color-coding to better visualize cell fusion.

We have added an unannotated version of the image to aid clarity.

213 "The NR emerges in the later half of embryogenesis": Earlier paper showing this should be cited here (Rapti et al., 2017, PMID: 28846083).

We have added this citation.

217 "even neurite topography through combinations of two-color imaging": Earlier studies with C. elegans two-color FM imaging of neurites (including in the embryo!) can be cited (Heiman, Shaham, 2009, PMID: 19344940, Rapti et al., 2017, PMID: 28846083, Moyle et al., 2020, PMID: 33627875).

We have attempted to clarify in the text that we are not referring to multi color imaging in general (though this is indeed a vital technique) but specifically referencing the idea of triangulating relative process placement, with the goal of highlighting the different resolutions of information about this available in EM and FM views of the same neurites in Figure 4b. Here we add a reference to Yip and Heiman 2018 pMID 30117807 as another relevant example of what we mean, (while using 3 colors in one pass) but do not feel that the other suggested references, which we cite elsewhere in more directly relevant contexts elsewhere in the manuscript, are apropos here.

220 "We scan through EM data to identify neurites, and backtrace them to soma": The authors should explain the exact approach used for process tracing (manual, semi-/ automatic, etc).

We clarify in Methods that we perform manual backtracing of neurites to soma to establish neurite identity.

240-250 Figure 4b-4e. Color-coding is not consistent between models, FM, EM images, i.e. sublateral pioneers are red in Figure 4b EM but light blue in Figure 4c, and Plim-4 which labels sublateral pioneers is red in Figure 4e. Consistent color-coding would help the reader.

See point 22 above for general comments.

Coloring in panel e was based on source strain, we agree that this is confusing given it is used only in this panel and have recolored data to match the color of panels c,d,h to minimize potential confusion. This is noted in the revised caption.

Figure 4b is in a distinct color scheme to highlight the grouping of the neurites in the EM at the level of the accompanying FM data. We have retained our original color scheme here.

Figure 4c, S5a It is unclear if the drawn arrows' length is proportional to the actual neurite total growth (across EM sections of the same embryo)? For example, what is the relative length of SIA/SMD and ASH, in the model (Figure S5a) and in the actual EM (Figure S5f,g)?

We clarify in the figure legend that the drawing reflects the approximate extent of neurite outgrowth

268-271 "comparing the left- and right-side behavior of the first cells to grow into the NR, namely SIADL/R and SMDDL/R, suggests strong symmetry in behavior. Both sides have broken out at the bean stage, with similar-sized outgrowths […] their tips meet precisely beneath the midpoint of the ALA […]".This is in line with previous studies, indicating such symmetries and ALA's position, these studies should be cited (Rapti et al., 2017, PMID: 28846083; Insley and Shaham, 2018, PMID: 29590193; Moyle et al., 2021, PMID: 33627875). In order to establish symmetry beyond previous observations, it appears key to providing quantitative data, calculating the length of bilateral processes by tracing them in EM series.

We agree the general notion of symmetry in the nervous system is not new. Our contribution regards the scale and resolution at which it remains symmetric. Our observation is, to our knowledge, the first documentation of developmental symmetry at this level of detail.

To add nuance to our observation, we have measured the lengths of the processes of early neurons (and potential functional pioneers) SIAD and SMDD, we have added a sentence referencing these results which suggest the outgrowth between the left and right is controlled to within a few percent of neurite length.

282 The first paper demonstrating this dynamic process should be cited (Heiman, Shaham, 2009, PMID: 19344940).

As mentioned above we have added this reference.

Figure S5B Shading of EM figure in Box b doesn't allow neurite visualization.

We have added unshaded versions of each panel in these supplemental figures to aid visualization.

Also (box B), how it is distinguished whether neurons extend neurites or have elongated cell bodies? It would help to provide the EM images with and without color-coding (and larger sizes) to facilitate visualization.

We have added a Methods section clarifying our treatment of neurite outgrowths, including how we make the distinction between elongated cell bodies and neurite outgrowth in EM. The distinction is easier in FM data where dynamic information informs interpretation of cell shape. The full EM data sets are provided for independent assessment as well as unshaded versions of panels as mentioned above.

274 "It is not clear […] if there is signaling to coordinate between the two sides."Some mechanisms related to the regulation of left-right symmetry of axon guidance are studied in Grossman, Giurumescu, Chisholm, 2013, PMID: 23979582. The paper should be cited.

We apologize for this oversight and have added this citation as an example of asymmetric guidance molecule influence.

Figure S3, S4, S5, S6, S7 are mislabeled, 2 figures are labeled S3 and legends do not correspond. Also, scale bars are missing in many supplementary figures and when present, there is no reference of scale bar size in the legend. These should be corrected accordingly.

We apologize for this error in numbering, and for the omission of scale bar dimensions in supplemental figure captions this has been corrected. All image panels have scale bars, in some cases shared between adjoining montage frames within a panel.

411 Some alignment of 1.5/2fold AT-EM data (available in webKnossos) is "jittery", may any additional image registration help eliminate this, for the better visualization of data?

We agree that the alignment of the AT data sets could be improved with additional manual annotation. We intend to improve this in future which will make for a more attractive visualization. Until then, the quality of slice alignment is sufficient to support the most critical tasks, browsing the stack, and following cell bodies and neurites. We have added some additional documentation of our alignment approach in methods, essentially automatic registration, with manual adjustment used only to correct qualitatively misaligned slices.

Reviewer #3 (Recommendations for the authors):The method is well-executed and thoroughly explained, and this manuscript shines in the temporal and spatial precision by which developmental processes are uncovered and described as a result. However much of these biological discoveries were buried in the supplementals. To create more relevance and excitement for this journal's audience, the authors should re-write parts of the manuscript, with a stronger emphasis on what the limitations of current imaging methods are, what cross-modality analyses improve upon, and particularly by bringing focus to the biological discoveries and putting them into the main article.

We thank the reviewer for their helpful suggestions and have attempted to contextualize our approach more in the text. We have also brought supplemental figures into the main article as secondary figures. Individual points are addressed in greater detail below.

In addition to the inherent challenges of EM-FM cross-comparison, in this article, the cross-modality alignment method overcomes two distinct issues: the challenges associated with manual annotation of embryo developmental EM series; as well as integration of spatial resolution of EM with the temporal resolution with FM. These concepts are briefly mentioned in the introduction and throughout the article, but do not become apparent to a non-specialist reader until the end. These current challenges should be further emphasized in the introduction when describing these imaging methods to highlight the benefits of the analysis method presented.

We have attempted to clarify these two distinct contributions in the introduction.

Note: supplemental figures are mis-numbered (there are two Figure S3s, and the following figures do not match figure captions). I will refer to supplemental figures as they are numbered in the figure captions.

Apologies for the mislabeling of supplemental figures, this has been corrected.

Most of Figures S5, S6 and S7 should be integrated into or added as additional main figures. These data beautifully highlight both the temporal and spatial resolution achieved by the cross-modality method used.

All supplemental figures have been moved into the main text as secondary figures associated with manuscript primary figures or placed in textual form in the Methods section.

How is "adjacency" defined between nuclei that are not physically touching? Are there specific parameters for what distance between cell nuclei counts as adjacent?

We have added an explicit pointer in the main text to the Methods section which describes how our adjacency graph is computed as the Gabriel Graph, a stable subset of Voronoi faces.

Line 236: what is meant by "breakouts observed in the EM at the comma stage span an hour?". Is this referring to lack of temporal resolution? This statement is confusing because Figure 4e and Table S4 referenced doesn't seem to show this to be the case.

We clarify in the text that we mean that cells observed as *having grown* out in the EM comma embryo span ~an hour in the FM timings of their initial outgrowth.